# LIMA1-alpha staining predicts curative intent surgery response in HPV negative head and neck cancer

Xi Qiao [1,13], Johannes Routila [2,13], Mari Tienhaara [3], Heikki Irjala [2],

Priyadharshini Parimelazhagan Santhi[1], Teemu Huusko [2], Linda Nissi[2], Ilkka Paatero [1],

Noora Lehtinen[4], Juha Rantala [4], Toni Viljanen[2], Ilmo Leivo[5,6], Petri Koivunen [7], Anna Jouppila-Mättö[8],

Rami Taulu[9], Leif Bäck[10], Tommy Wilkman [11], Eeva Haapio [2], Ilpo Kinnunen[2], Kari Kurppa [3],

Jukka Westermarck [1,12,14✉] & Sami Ventelä [1,2,14✉]

## Abstract

In many solid cancer types, surgery alone could be a sufficient first therapy option for a significant number of cancer patients. However, there are currently no diagnostic solutions to identify patients who could be stratified to surgery alone. To identify a biomarker predicting cancer surgery response, candidate biomarkers were studied in a non-metastatic head and neck squamous cell carcinoma (nmHNSCC) cohort well representative of the HPV-negative patient population. LIMA1 immunohistochemistry (IHC) with specificity-validated antibodies outperformed all other biomarkers in multivariable survival analyses of patients with nmHNSCC ($n = 128$, HR 2.10, $P = 0.006$). The prognostic effect was selective to LIMA1-alpha isoform IHC detection in patients who had received surgical therapy ($n = 184$, HR 2.39, $P > 0.001$). Strikingly, our real-world validation results, using two prospectively collected cohorts ($n = 15$ and $n = 86$), demonstrate that none of the LIMA1 negative patients died of HNSCC during the follow-up. Collectively, we report here the discovery of a diagnostic LIMA1-alpha IHC assay for HPV-negative HNSCC patient stratification to surgery-only therapy. Application of LIMA1 detection in routine nmHNSCC diagnostics would revolutionize the clinical management of HNSCC patients.

**Keywords** EPLIN; Disease-Specific Survival; Population-validated Tissue Microarray; PV-TMA
**Subject Categories** Biomarkers; Cancer; Methods & Resources

## Introduction

Across all solid cancer types, surgery is still the mainstay first-line therapy option for most of the patients. It is also known that in many cancer types, some patients with non-metastatic tumors of small size (T1-2, N0, M0) could achieve clinically comparable treatment outcomes with surgery-only as compared to a combination of surgery with adjuvant therapy. However, identification of patients that are suitable for surgery-only therapy, would require the development of a diagnostic biomarker identifying patients who do not develop disseminated disease after surgical removal of non-metastatic tumor at the time of diagnosis. Such patient stratification marker does not currently exist but is discovered here for the use in head and neck squamous cell carcinomas (HNSCC).

Head and neck cancers are the seventh most common malignancies worldwide (Sung et al, 2021). Approximately 90% of head and neck cancers are squamous cell carcinomas, which have a relatively high, ~50% 5-year mortality rate (Grégoire et al, 2010; Chow, 2020). In HNSCC, HPV-positive and HPV-negative disease have distinct etiology, pathology, immune landscape and patient outcomes as well as subsite prevalence. HPV-positive HNSCCs are almost exclusively derived from the oropharynx and their HPV positivity indicates for better patient survival (Chow, 2020). Recurrence and metastasis are the main reasons for poor survival and mortality among HNSCC patients (Chow, 2020; Bozec et al, 2019). Importantly, no new diagnostic or therapeutic methods that would have markedly improved the survival of HNSCC patients have been introduced for decades (Chow, 2020). One of the most challenging shortcomings in the implementation of personalized cancer therapy in HNSCC, is the scarcity of clinically approved biomarkers. Therefore, HNSCC biomarker studies done with high-quality standards are clearly needed to identify clinically translatable biomarkers for patient therapy stratification.

[1]Turku Bioscience Centre, University of Turku and Åbo Akademi University, Tykistökatu 6, Turku 20520, Finland. [2]Department for Otorhinolaryngology, Head and Neck Surgery, University of Turku and Turku University Hospital, Kiinamyllynkatu 4-8, Turku 20521, Finland. [3]Institute of Biomedicine, and MediCity Research Laboratories, University of Turku, Turku 20520, Finland. [4]Misvik Biology Ltd, Karjakatu 35 B, Turku 20520, Finland. [5]Department of Pathology, Turku University Hospital, Kiinamyllynkatu 4-8, Turku 20521, Finland. [6]Institute of Biomedicine, Pathology, University of Turku, Turku 20520, Finland. [7]Department of Otorhinolaryngology, Head and Neck Surgery, University of Oulu and Oulu University Hospital, Oulu, Finland. [8]Department of Otorhinolaryngology, Head and Neck Surgery, Kuopio University Hospital, Kuopio, Finland. [9]Department of Otorhinolaryngology, Head and Neck Surgery, Tampere University Hospital and University of Tampere, Tampere, Finland. [10]Department of Otorhinolaryngology, Head and Neck Surgery, University of Helsinki and Helsinki University Hospital, Helsinki, Finland. [11]Department of Oral and Maxillofacial Diseases, University of Helsinki and Helsinki University Hospital, Helsinki, Finland. [12]InFLAMES Research Flagship Center, University of Turku, Turku, Finland. [13]These authors contributed equally: Xi Qiao, Johannes Routila. [14]These authors contributed equally: Jukka Westermarck, Sami Ventelä. ✉E-mail: jukwes@utu.fi; satuve@utu.fi

Currently, the only biomarkers that have been approved for wider clinical use in HNSCC are the HPV surrogate marker p16 and programmed death-ligand 1 (PD-L1) (Chow, 2020). However, both of these biomarkers have limitations in clinical practice, as p16 assay does not justify therapy stratification, and PD-L1-targeting immunotherapies are applicable to only a small proportion of HNSCC patients with advanced stage of the cancer (Chow, 2020; Bozec et al, 2019; Gillison et al, 2019; Mylly et al, 2022; Burtness et al, 2019). A particularly important clinical challenge in HNSCC is how to determine the aggressive cancer behavior in patients having non-metastasized HNSCC (nmHNSCC) at the time of diagnosis. These patients are typically offered monotherapy, most commonly surgery, as a first-line treatment option. However, a significant proportion of these patients develop either local recurrences or distant metastases during follow-up. Finding a biomarker that would reliably predict the metastatic propensity of nmHNSCC tumors would enable individualized cancer treatment planning, based on which those patients who could be cured by surgery-only could be spared from the serious side effects of multimodal therapies.

In this study, we discover and validate immunohistochemical detection of LIM Domain And Actin Binding 1 (LIMA1) protein as a novel diagnostic approach to identify those HNSCC patients that could be selected for curative intent surgical treatment. Using validated specific antibodies and several independent clinical cohorts, including two prospectively collected diagnostic trials, the results of this study, reported following the REMARK criteria (Sauerbrei et al, 2018) provide strong clinical proof-of-concept for development of LIMA1 IHC detection towards clinical utility for detecting HNSCC patients benefitting of cancer surgery. Further, we provide a functional rationale why high LIMA1 expression is linked with significantly shorter disease-free survival of HNSCC patients by demonstrating that high LIMA1 expression promotes metastatic capacity of patient-derived HNSCC cells in vivo.

# Results

## Validation of representativeness of the patient cohort and specificity of the antibodies used for immunohistochemistry

Our unique research strategy to tackle low reproducibility and translatability of HNSCC biomarker research (Ren et al, 2020) has been to use patient cohorts, that have been robustly validated to be representative of an average patient population, and then validate the findings in multiple independent cohorts with specificity-validated antibodies (Mylly et al, 2022; Routila et al, 2021; Punovuori et al, 2024; Nissi et al, 2025). This research strategy was used here to address whether LIMA1 could function as a diagnostic marker to identify HNSCC patients suitable for surgery-only therapy. From the original HNSCC cohort ($n = 476$) (Routila et al, 2021), we identified 312 HNSCC cases that did not have any signs of HNSCC metastasis at the time of diagnosis and could therefore considered for surgery-only therapy. Out of these non-metastatic HNSCC (nmHNSCC) cases, both tissue samples and clinical details were available for 128 patients (Fig. 1A). Notably, most patients in the cohort were from other tumor sites than Oropharynx that is the site from where most of the HPV-positive cancers are derived (Table 1). The representative nature of this

nmHNSCC PV-TMA cohort (cohort 1) was validated by demonstrating that the cohort did not differ statistically significantly from the overall nmHNSCC patient population by any clinically meaningful criteria (Table 1). Notably, 25% ($n = 32$) of the patients with nmHNSCC died of HNSCC cancer during 5 years of follow-up (Table 1). This indicates that although these patients did not display any sign of metastasis at the time of diagnosis, a significant proportion of the nmHNSCCs develop towards metastatic disease during the follow-up. Identification of those patients at the time of surgery by a biomarker compatible with routine hospital diagnostics would be a transformative advance in HNSCC management.

Antibody specificity is an absolute requirement for IHC-based diagnostics, but many antibodies are unspecific, and many biomarker studies lack proper antibody specificity validation (Ren et al, 2020; Lund-Johansen, 2023). Therefore, we robustly validated the specificity of the three LIMA1 antibodies used in this study. The antibodies used were either commercial polyclonal rabbit antibody (HPA023871), monoclonal mouse antibody (SC-136399) or custom-made polyclonal antibody (RB581). Whereas HPA023871 and SC-136399 were raised against epitope common for two known LIMA1 isoforms, LIMA1-alpha and -beta (Collins et al, 2015), the RB581 was raised by us against LIMA1-beta-specific epitope (Fig. EV1A). The LIMA1 isoforms are expressed from alternative promoter regions, resulting in expression LIMA1-alpha (600 amino acids) and LIMA1-beta (760 amino acids) (Collins et al, 2015). The specificities of the antibodies were verified by three-tiered analysis, including (a) demonstration that they only detect the expected size proteins without additional bands in the entire Western blot membrane, (b) loss of this signal in Western blot and c) loss of signal in immunofluorescence staining analysis by siRNA-mediated protein knock-down. As shown in Fig. EV1, all three antibodies fulfilled these quality requirements to be used for LIMA1 IHC analysis from HNSCC tissues.

## Low LIMA1 predicts for over 50% 5-year survival among nmHNSCC patients

Importantly, upon IHC analysis by HPA023871 antibody, LIMA1 expression had a strong prognostic impact on overall survival (OS) in nmHNSCC based on univariate analysis ($P = 0.006$) (Fig. 1B). Whereas 50% of patients with LIMA1 high tumor succumbed in less than 3 years, the patients with LIMA1 low tumor did not reach the 50% mortality even after 5-year follow-up (Fig. 1B, red dashed line). LIMA1 remained an independent poor prognosis factor also in multivariable survival analysis, when a prognostic model including patient age, high T class, and alcohol consumption was used (Fig. 1C; Appendix Fig. S1A). As T class is the most important prognostic measure in routine clinical HNSCC diagnostics, it is important to observe that LIMA1 remained a highly significant prognostic factor despite the inclusion high T class in the prognostic model (LIMA1's HR 2.10; 95% CI 1.24–3.58) (Fig. 1C). Importantly, demonstrating uniqueness of LIMA1 in nmHNSCC prognostification, expression levels of EGFR, p16, OCT4, MET, TP53, NDFIP1, or CIP2A did not offer statistically significant prognostic resolution from the same nmHNSCC cohort (Fig. 1C; Appendix Figs. S1 and S2 and Appendix Table S1). Association between low LIMA1 mRNA expression and better HNSCC patient overall survival was also confirmed independently from the TCGA dataset (HNSCC cohort 2), in which the prognostic effect of low LIMA remained highly significant in the multivariable model controlling for age, gender, T class, and nodal status (HR 0.53; 95%

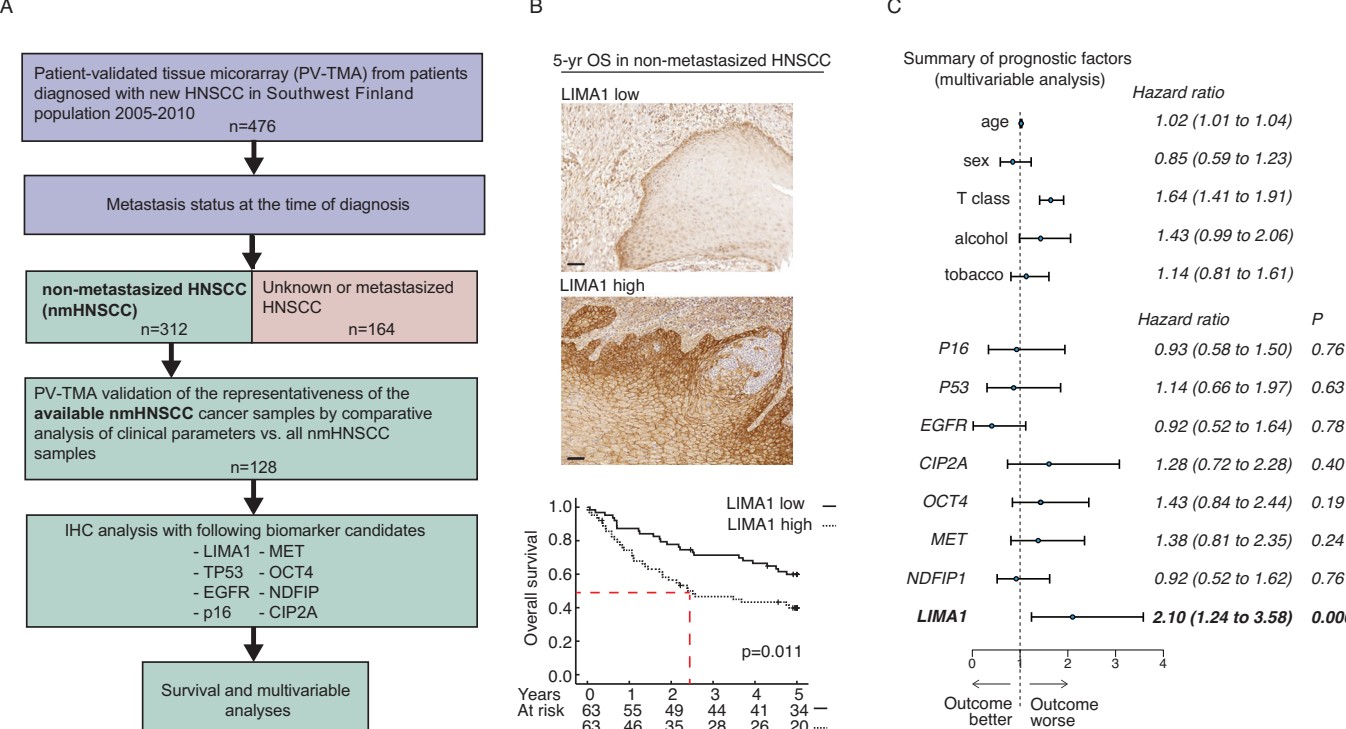

**Figure 1. Low LIMA1 expression in surgical nmHNSCC sample is an independent biomarker for favorable prognosis.**

(A) Description of the clinical cohort that was used in the formation of an unbiased and population-validated HNSCC tissue microarray (PV-TMA) and for identification of non-metastasized (nmHNSCC) HNSCC patients ($n = 128$). Patient samples were immunostained with LIMA1-specific antibody. The same PV-TMA was stained additionally for TP53, EGFR, p16, CIP2A, MET, and OCT4 (Appendix Figs. S1 and S2). (B) Representative immunohistochemical stains of LIMA1 LOW and LIMA1 HIGH HNSCC samples. High LIMA1 expression strongly associated with poor 5-year overall survival in the nmHNSCC PV-TMA patient cohort. Log-rank method was used for determining the significance of the difference for survival distributions. Scale bar: 100 μm. The exact *P* values were as indicated. (C) Multivariable clinical prognostic model including age, high T class, and alcohol consumption ($n = 312$). The test demonstrated strong survival impact for LIMA1 immunohistochemistry that is independent of any other analyzed factors.

CI: 0.40–0.69, $P < 0.001$, Fig. EV2A). The TCGA data were also used to compare the expression of LIMA1 mRNA between HPV-positive, HPV-negative, and normal control samples. Consistent with both LIMA1 and HPV negativity being associated with more aggressive HNSCC, LIMA1 expression was statistically higher in HPV-negative HNSCC samples (Appendix Fig. S3A). Notably, indicating that the prognostic role for LIMA1 could be extended at least to some other cancer types, statistically significant reduction of OS was observed also in TCGA cohort of pancreatic cancer treated with curatively intended surgery ($P = 0.009$, Fig. EV2B).

These results indicate that high LIMA1 expression is associated with poor prognosis in HNSCC, and that LIMA1 IHC could provide a new diagnostic approach for the identification of those nmHNSCC patients with favorable surgical therapy response. Furthermore, the prediction of surgical response by LIMA1 expression does not seem to be limited to HNSCC cancers.

## LIMA1α tumor expression predicts poor survival among surgically treated HNSCC patients

LIMA1 gene has two alternative active promoter regions resulting in expression of two different-sized LIMA1 protein isoforms, LIMA1-alpha (600 amino acids) and LIMA1-beta (760 amino

acids) (Collins et al, 2015) (Fig EV1A). However, whether these two isoforms have biologically and pathologically different roles in HNSCC is currently unrevealed.

To gain insights whether the known LIMA1 isoforms, alfa and beta (Collins et al, 2015), might have different prognostic and pathological roles in HNSCC, we performed exon-level mRNA expression analyses of LIMA1 exons in HPV-positive and HPV-negative samples in the HNSCC TCGA dataset. In these results, LIMA1-beta isoform (exons 1–3) did not associate with HPV status (Appendix Fig. S3B,C). However, HPV-negative samples showed a significant correlation with LIMA1-alpha (exons starting from exon 4), supporting the hypothesis that LIMA1-alpha is associated with a worse prognosis in HNSCC. To further investigate the role of LIMA1 isoforms in HNSCC cancer, we collected a prospective HNSCC patient tumor tissue cohort, from which the expression of LIMA1 isoforms was studied by Western blotting with SC-136399 antibody (WB, Fig. 2A). In line with the focus of this work, the 15 cancer patients included in this prospective cohort were newly diagnosed HNSCC patients who were treated with curative intent surgery (HNSCC cohort 3). The characteristics of the patients in the prospective follow-up study are shown in Appendix Table S2. Notably, the LIMA1-positive HNSCC tissue samples expressed predominantly the shorter LIMA1-alpha isoform (WB, Fig. 2A).

**Table 1.** Validation of representativeness of the nmHNSCC tissue microarray (TMA) univariate (left panels) and multivariable (right panels) analysis of nmHNSCC TMA ($n = 128$) inclusion bias as compared to entire nmHNSCC population ($n = 312$).

| | Total | | TMA patients | | Univariate | | Multivariable | |
|---|---|---|---|---|---|---|---|---|
| | $n = 312$ | % | $n = 128$ | % | OR (95% CI) | P | OR (95% CI) | P |
| **Gender** | | | | | | | | |
| Male | 211 | 68% | 79 | 62% | 0.64 (0.39–1.03) | 0.064 | 0.54 (0.31–0.93) | 0.027 |
| Female | 101 | 32% | 49 | 38% | 1 | – | 1 | – |
| **Age at diagnosis** | | | | | | | | |
| <65 | 132 | 42% | 58 | 45% | 1.23 (0.78–1.94) | 0.37 | Not included | |
| >65 | 180 | 58% | 70 | 55% | 1 | – | | |
| **Smoker** | | | | | | | | |
| >20 pack yrs | 170 | 54% | 76 | 59% | 1 | – | | |
| <20 pack yrs | 142 | 46% | 52 | 41% | 1.40 (0.89–2.21) | 0.15 | Not included | |
| **Alcohol consumption** | | | | | | | | |
| Yes | 82 | 26% | 36 | 28% | 1 | – | | |
| No | 230 | 74% | 92 | 72% | 0.85 (0.51–1.42) | 0.54 | Not included | |
| **Primary tumor site** | | | | | | | | |
| Oral cavity | 176 | 56% | 73 | 57% | 1 | – | 1 | – |
| Oropharynx | 26 | 8% | 17 | 13% | 2.67 (1.13–6.31) | 0.026 | 3.42 (1.38–8.51) | 0.008 |
| Larynx | 87 | 28% | 28 | 22% | 0.67 (0.39–1.15) | 0.15 | 0.85 (0.45–1.59) | 0.61 |
| Hypopharynx | 10 | 3% | 4 | 3% | 0.94 (0.26–3.45) | 0.93 | 1.11 (0.28–4.39) | 0.88 |
| Other | 13 | 4% | 6 | 5% | 1.21 (0.39–3.75) | 0.74 | 1.55 (0.48–5.01) | 0.47 |
| **T class** | | | | | | | | |
| T0-2 | 232 | 74% | 92 | 72% | 1 | – | | |
| T3-4 | 80 | 26% | 36 | 28% | 1.25 (0.75–2.08) | 0.40 | Not included | |
| **Recidive in 5 yrs** | | | | | | | | |
| Yes | 86 | 28% | 38 | 30% | 1.27 (0.76–2.11) | 0.36 | Not included | |
| No | 208 | 67% | 80 | 63% | 1 | – | | |
| No curative treatment | 18 | 6% | 10 | 8% | 2.00 (0.76–5.28) | 0.16 | Not included | |
| **Living at 5 yrs** | | | | | | | | |
| Yes | 180 | 58% | 64 | 50% | 1 | – | 1 | – |
| No, died of HNSCC | 73 | 23% | 32 | 25% | 1.74 (0.98–3.08) | 0.058 | 1.88 (1.03– 3.45) | 0.041 |
| No, died of another cause | 59 | 19% | 32 | 25% | 2.38 (1.39– 4.07) | 0.002 | 3.08 (1.73–5.48) | <0.001 |
| **Surgical treatment** | | | | | | | | |
| Local operation | 221 | 71% | 97 | 76% | 1.57 (0.94–2.61) | 0.087 | NS | – |
| Neck dissection | 64 | 21% | 33 | 26% | 0.58 (0.33–1.00) | 0.052 | 1.79 (0.97–3.30) | 0.064 |
| No surgery | 90 | 29% | 30 | 23% | 1 | – | 1 | – |

Results from logistic regression modeling.
NS not significant.

We also studied the isoform expression across 14 patient-derived HNSCC cell lines. Indeed, in all these cell lines the LIMA1-alpha was the predominantly expressed isoform (Fig. EV1B). This was however not the case across 13 triple-negative breast cancer cell lines that showed a mixed pattern of LIMA1-alpha or -beta dominance (Fig. EV1C). As LIMA1 positivity in IHC and WB analyses by SC-136399 was concordant in each case in this prospective study (Fig. 2A), we conclude from this data that LIMA1-alpha is the primary isoform of LIMA1 expressed in HNSCC tumors.

Two years after the primary surgery, each patient´s follow-up data in this HNSCC cohort was evaluated and combined with LIMA1 status analyzed by WB from the patient's primary tumor. Strikingly, all the HNSCC patients who developed metastasis, had high LIMA1-alpha tumor expression upon primary surgery (Fig. 2B,C), whereas none of the five patients with LIMA1-alpha low carcinoma at diagnosis developed HNSCC metastasis during the 2-year follow-up (Fig. 2B,C). Further, the HNSCC-related mortality associated very strongly with LIMA1-alpha expression upon original diagnosis, as there were no HNSCC-related deaths in

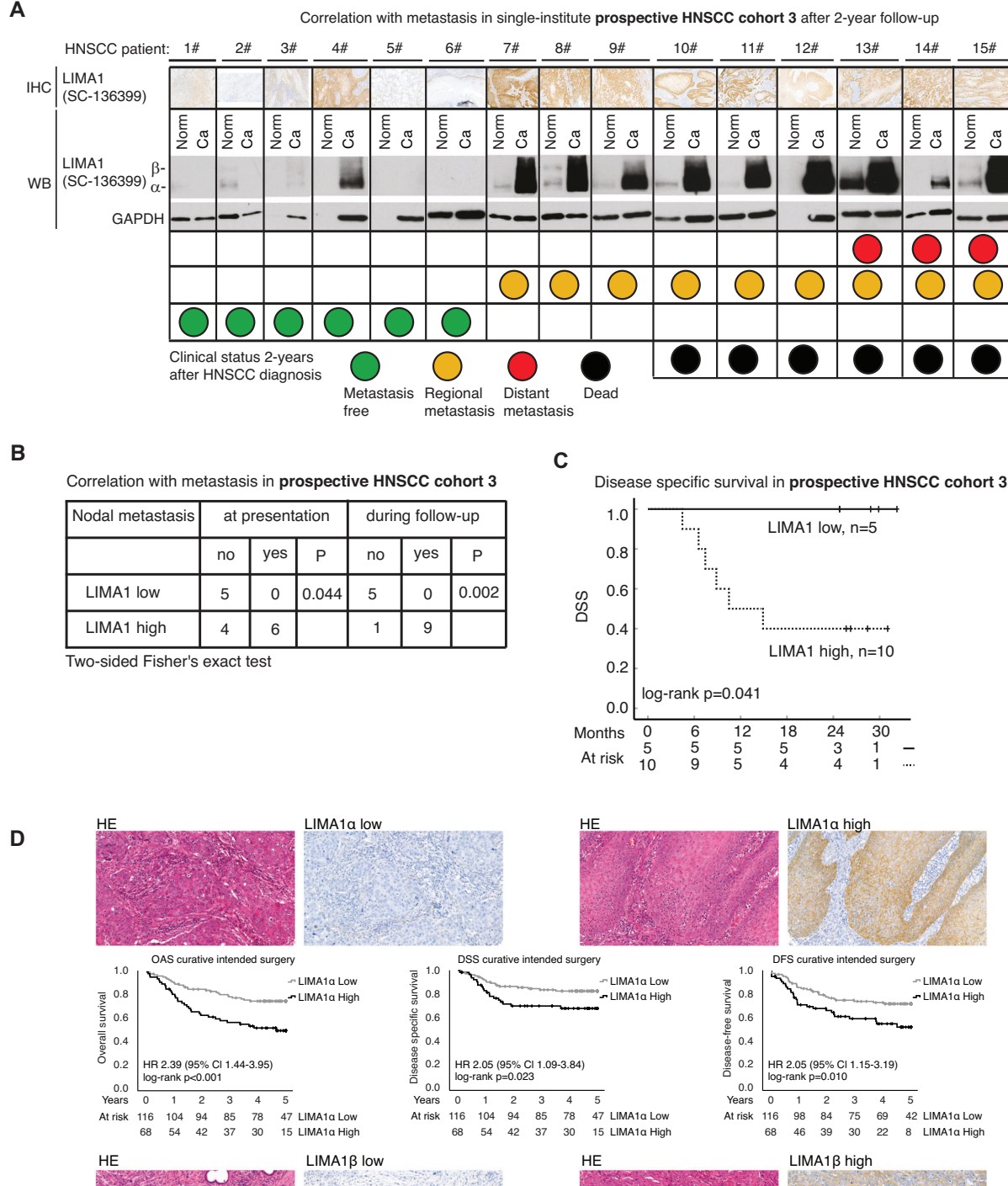

**Figure 2. LIMA1-alpha IHC detection identifies HNCSS patients that will have favorable response to curative intent surgery.**

(A) A prospective 2-year follow-up study of 15 randomly selected patients diagnosed with new HNSCC and underwent with curative intended surgery (HNSCC cohort 3). LIMA1 immunohistochemical (IHC) staining and western blot analysis (both with sc-136399) were performed on the carcinoma samples (ca). Moreover, LIMA1 western blot analyses were performed from a fresh tissue sample taken outside the tumor area (Norm). A traffic light model that considers HNSCC patient survival as well as HNSCC cancer metastasis was created as follows: patients who were alive 2 years after surgery received a green light, patients who had regional lymph node metastases, received a yellow light, patients with distant metastases received a red light and the patients who died during the follow-up were marked with a black. (B) LIMA1 positivity was significantly associated with the occurrence of nodal metastasis both at presentation and during follow-up. Two-sided Fisher's exact test was used for the statistical analysis. The exact P values were as indicated. (C) LIMA1 positivity was remarkably associated with poor survival during follow-up. (D) Shown are representative H&E, LIMA1-alpha (sc-136399), LIMA1-beta (RB581) Low and High IHC stainings. Overall survival (OAS), disease-specific survival (DSS) and disease-free survival (DFS) were analyzed from the oral cavity HNSCC cohort 4 (n = 185; Table 2). Exact P values were as indicated. Source data are available online for this figure.

the LIMA1-alpha low cohort, whereas all patients who died during follow-up had LIMA1-alpha high primary tumor (Fig. 2B,C).

## Comparison of the prognostic role of LIMA1a and LIMA1b in HNSCC

Results above strongly indicate that HNSCC patients with high LIMA1-alpha is associated with more aggressive HPV-negative HNSCC whereas low LIMA1-alpha tumor expression upon diagnosis predicts for a significantly lower likelihood to develop a deadly metastatic disease, and therefore these patients would benefit particularly well from the cancer surgery. To validate these provocative results, we developed a LIMA1-beta specific antibody RB581 (Fig. EV1A,E,H). Thereafter, we performed a comparative IHC staining study using LIMA1 antibody sc-136399, detecting both isoforms, and LIMA1-beta selective RB581 antibody (Fig. EV1A,D–H). To further increase the clinical relevance and applicability of our results, this isoform comparison was done by using yet another retrospective HNSCC cohort, containing 185 oral cavity HNSCC patients treated with curative intended surgery as a first-line treatment option (HNSCC cohort 4) (Table 2). Reassuringly, cancers containing high positivity with LIMA1-alpha detecting antibody sc-136399 (n = 68, 62.7%), showed a statistically significant decrease in overall survival, disease-specific survival, and disease-free survival (Fig. 2D). Patients with high LIMA1 based on sc-136399 IHC had only 22% 5-year disease-specific survival, whereas patients with low LIMA1 had a survival of 41% in this independent cohort of HNSCC patients treated with primary cancer surgery. However, when this cohort was stained with the LIMA1-beta selective antibody RB581, the predictive value of LIMA1-beta positivity was clearly lesser than LIMA1-alpha positivity (Fig. 2D).

Collectively, the results from three independent cohorts of HNSCC patients treated with intended curative surgery demonstrate, that LIMA1 IHC (OS in cohort 1: P = 0.006, cohort 3: P = 0.041, cohort 4: P < 0.001) could be a clinical practice-changing diagnostic approach for identification of patients that will have favorable response to curative intent surgery. Furthermore, our results indicate that while both LIMA1 isoforms may be involved in the aggressiveness of HNSCC, diagnostically LIMA1-alpha detection is superior over LIMA1 beta in differentiating the patients based on their disease-specific survival.

## A real-world demonstration of the performance of LIMA1 as a predictive biomarker in a prospective national multicenter HNSCC study

The results above indicate for a clinical scenario, where low LIMA1 expression could be used as a biomarker to identify HNSCC

patients eligible for surgery-only first-line therapy. Although actual therapy de-escalation trial to demonstrate such utility of LIMA1 biomarker is under progress to be initiated, we modeled this scenario by launching a national prospective multicenter HNSCC biomarker study across every University hospital in Finland (Turku, Helsinki, Tampere, Oulu, and Kuopio). This study thus covers the total Finland's population and is void of socioeconomic or treatment center biases. Altogether 95 patients were recruited with following inclusion criteria: (1) patients were diagnosed with new HNSCC, and (2) treatment was initiated with curative intent surgery (Fig. 3A). More than 90% of patients in this cohort (HNSCC cohort 5) had a small, low T class (pT1-2) tumor with a good prognosis at baseline (Table 3). Therefore, this cohort provided a perfect prospective study material to challenge performance of LIMA1 IHC assay to identify patients that had an aggressive cancer besides traditionally favorable prognostic features (Table 3).

After 2-year follow-up, the surgically removed HNSCC primary cancer samples were IHC stained with LIMA1 antibodies sc-136399 and RB581 (Fig. 3B,C,E). Whereas the final results of this prospective multicenter study will be reported after 5-year follow-up data from all patients is available, we here report an interim analysis of data with 86 patients and 2-year follow-up. On analysis of LIMA1 (alpha by sc-136399) positivity, the LIMA1 low and LIMA1 high patients represented almost equally sized cohorts (LIMA1-low 51% and LIMA1-high 49%). Remarkably, analyses of disease-specific survival of these 86 patients revealed that none of the LIMA1 (alpha) low patients died of HNSCC during follow-up (Fig. 3D), whereas all HNSCC-related deaths during the follow-up occurred in LIMA1-positive HNSCC patients. Supporting our results above related to the relevance of isoform-specific LIMA1 detection, high LIMA1-beta expression was less significantly associated with poor HNSCC patient survival (Fig. 3E,F).

Combined with results from the first prospective study (Fig. 2C), these real-world clinical studies reveal a striking performance of negative LIMA1 IHC staining in recognizing patients that will not die of HNSCC during a clinically relevant follow-up period. Based on stringent antibody specificity validation, repeatability of the results across independent cohorts, and compliance with REMARK criteria on reporting, these results position LIMA-alpha detection by IHC as a novel clinically translatable HNSCC biomarker. Overall, in our patient cohorts 1,3,4,5, only 5.9% (n = 25) of patients had p16-positive HNSCC (Appendix Table S3), indicating that our results apply particularly to HPV-negative HNSCC and that LIMA1-alpha detection can identify patients with significantly different clinical outcomes from the HPV-negative cases.

**A**

Finnish multicenter national **prospective HNSCC cohort 5**

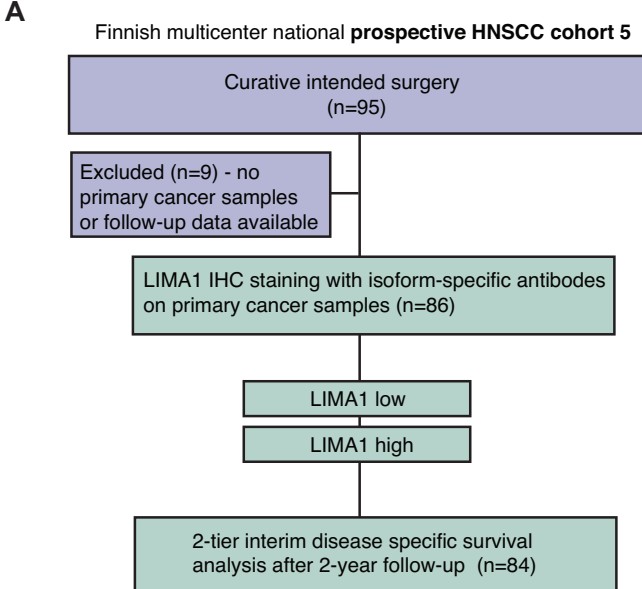

**B**

LIMA1 negative/low

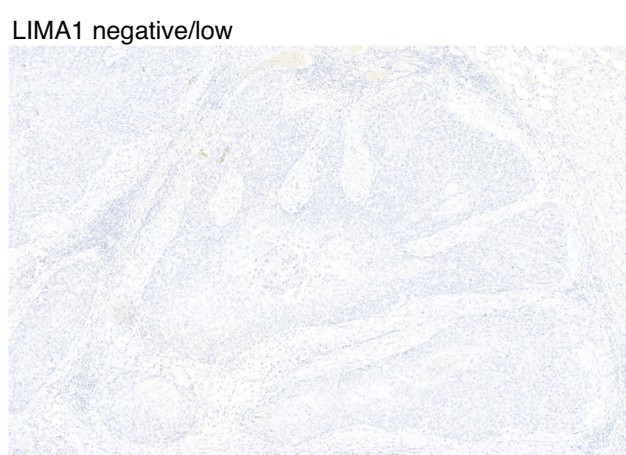

**C**

LIMA1 high

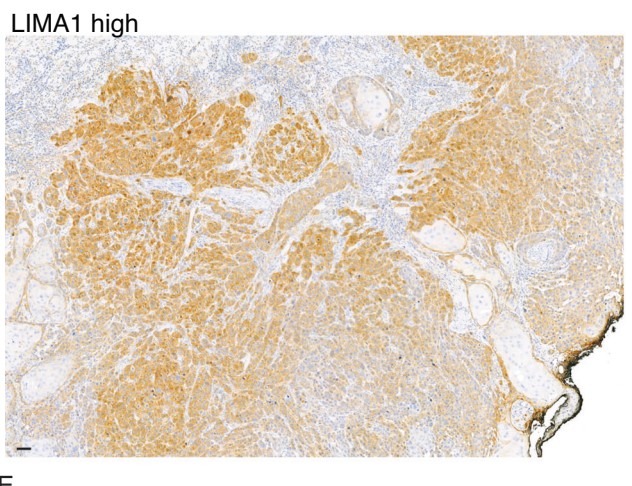

**D**

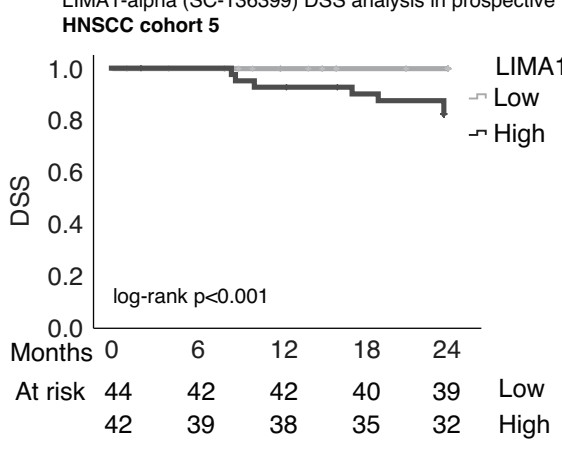

**E**

Representative IHC stains of LIMA1-beta specific antibody (RB581)

LIMA1 low LIMA1 high

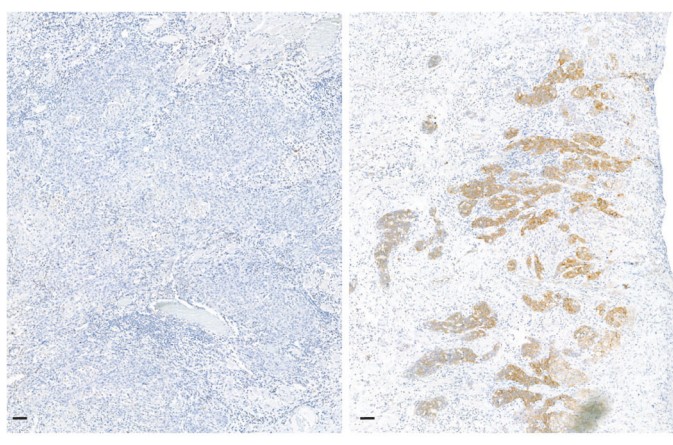

**F**

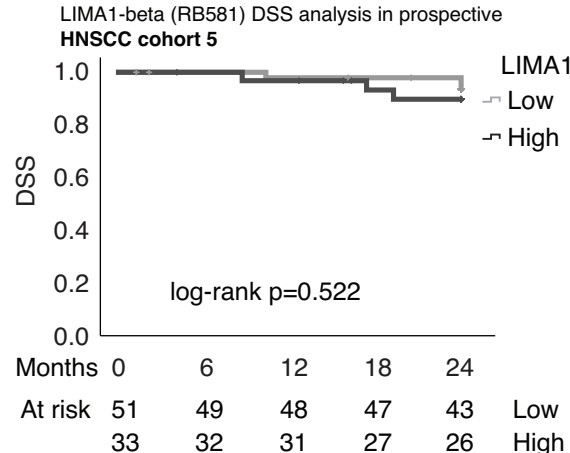

**Figure 3.   A real-world demonstration of performance of LIMA1 as a predictive HNSCC biomarker.**

(A) Flowchart representation of the Finnish multicenter prospective HNSCC study protocol. Altogether, 95 patients with new HNSCC and curative intended surgery were recruited to the study. LIMA1 IHC staining was performed with two isoform-specific antibodies. Two-tier interpretation was used in the analysis of staining intensities. (B, C) Representative LIMA1 IHC stains. (D) Disease-specific survival data (DSS) with 2-tire LIMA1 staining results with sc-136399 antibody. (E, F) Representative LIMA1 IHC stainings (with RB581 antibody) and disease-specific survival (DSS) data with LIMA1-beta-specific antibody. All scale bars indicated are 100 µm.

**Table 2.   Validation cohort of oral cavity squamous cell carcinoma patients treated with curative intended surgery ($n = 185$).**

| | Total $n = 185$ | % |
|---|---|---|
| **Gender** | | |
| Male | 92 | 49.7 |
| Female | 93 | 50.3 |
| **Age at diagnosis** | | |
| <65 | 64 | 34.6 |
| ≥65 | 121 | 65.4 |
| **Smoker** | | |
| <20 | 97 | 52.4 |
| ≥20 | 88 | 47.6 |
| **Alcohol consumption** | | |
| No | 150 | 81.1 |
| Yes | 35 | 18.9 |
| **T class** | | |
| T0-2 | 133 | 71.9 |
| T3-4 | 52 | 28.1 |
| **N class** | | |
| N0 | 128 | 69.2 |
| N+ | 57 | 30.8 |
| **Recidive in 5 years** | | |
| Yes | 59 | 31.9 |
| No | 122 | 65.9 |
| No curative treatment | 4 | 2.2 |
| **Living 5 years** | | |
| Yes | 123 | 33.5 |
| No | 62 | 66.5 |
| **LIMA1 IHC** | | |
| LIMA1-alpha low | 116 | 62.7 |
| LIMA1-alpha high | 68 | 36.8 |
| LIMA1-beta low | 123 | 66.5 |
| LIMA1-beta high | 57 | 30.8 |
| **p16 status** | | |
| Positive | 8 | 4.3 |
| Negative | 175 | 94.6 |
| Data missing | 2 | 1.1 |

## LIMA1 regulates HNSCC invasion in vitro and metastases in vivo

In HNSCC, disease relapse and disease-specific poor survival are very intimately associated with disease metastasis. Therefore, we investigated the functional role of LIMA1 in HNSCC cell invasion as a potential mechanistic explanation for its clinical association with disease-specific deaths. LIMA1 knockdown by siRNA decreased migration of HNSCC cells in the wound-healing assay (Fig. EV3; Appendix Fig. S4A,B). In the inverted 3D invasion assay (Jacquemet et al, 2016), UT-SCC14, UT-SCC45, and UT-SCC60B cell lines showed the highest invasion ability and were therefore selected to LIMA1 depletion experiments (Appendix Fig. S4C). Statistically significant downregulation in 3D invasion capability was demonstrated in all tested HNSCC cell lines across five independent experiments (Appendix Fig. S4D).

To study the role of LIMA1 in cancer invasion and metastasis formation in vivo, LIMA1 was depleted from one of the most invasive HNSCC cell line, UT-SCC14, and the cells were applied to a zebrafish embryo xenograft model (Appendix Fig. S4E). In this model, LIMA1 silencing did not impact tumor size, but the number of metastasizing cells decreased significantly (Fig. 4A–C). To ensure that the positive role for LIMA1 in HNSCC cell metastasis was not limited to established cancer cell lines, zebrafish xenograft (PDX) experiments were repeated with HNSCC patient-derived xenografts (PDX) (Appendix Fig. S5A,B). LIMA1 silencing from a PDX derived from HNSCC#16 tumor with high LIMA1-alpha expression (Appendix Fig. S5A; Fig. 4D) did not impact tumor size (Fig. 4E), while the number of invading cells in the zebrafish embryo xenograft model was statistically significantly decreased (Fig. 4F). In a reciprocal LIMA1 experiment, we used a PDX derived from a HNSCC patient with LIMA1 low tumor (Appendix Fig. S5B; Fig. 4G, HNSCC#17) and conditionally overexpressing LIMA1 from a lentiviral Tet-inducible gene expression vector. LIMA1 expression upon Doxycycline treatment was ensured by western blot (Fig. 4G). Fully supportive of the gain-off results above, overexpression of LIMA1 on HNSCC#17 PDX did not alter tumor size (Fig. 4H) but increased the number of metastasizing HNSCC cells in the zebrafish model (Fig. 4I). In addition, these functional results perfectly support the biomarker discovery results above, as HNSCC16# patient with LIMA1-alpha high tumor died within a year of widespread HNSCC, while HNSCC17# patient with LIMA1-alpha low tumor is still alive more than 3 years after HNSCC diagnosis.

LIMA1 has been implicated in the regulation of epithelial–mesenchymal transition (EMT), which is one of the most important phenomena that worsens HNSCC prognosis (Ma et al, 2022; Punovuori et al, 2024). Therefore, protein levels of a classic EMT marker vimentin was analyzed after LIMA1 silencing in HNSCC cells (Fig. 4J–L; Appendix Fig. S5C–F). In all tested cell lines and patient-derived HNSCC16#, LIMA1 downregulation using siRNA resulted in the reduction in vimentin levels. Furthermore, overexpression of LIMA1 in HNSCC17# cells robustly increased vimentin levels (Fig. 4L), suggesting that LIMA1 expression alone is sufficient to promote EMT phenotype. To provide an independent validation to these results, we used TCGA HNSCC dataset to analyze genes co-expressed with LIMA1 (Fig. EV4A). By the gene set enrichment analysis (GSEA), EMT

**Table 3.  Finnish national multicenter prospective HNSCC cohort (*n* = 86).**

|  | Analyzed | |
| --- | --- | --- |
|  | *n* | % |
|  | 86 | 90 |
| **Gender** | | |
| Male | 43 | 50 |
| Female | 43 | 50 |
| **Age at diagnosis** | | |
| <65 | 37 | 43 |
| ≥65 | 49 | 57 |
| **T class** | | |
| T1-T2 | 81 | 94 |
| T3-T4 | 5 | 6 |
| **pT class** | | |
| pT1 | 37 | 43 |
| pT2 | 44 | 51 |
| pT3 | 5 | 6 |
| **pN class** | | |
| N0-N1 | 79 | 92 |
| N2-N3 | 7 | 8 |
| **Primary tumor site** | | |
| Oral cavity | 86 | 100 |
| **Treatment** | | |
| **Primary tumor surgery** | | |
| Yes | 86 | 100 |
| No | 0 | 0 |
| **p16 status** | | |
| Positive | 4 | 4 |
| Negative | 38 | 40 |
| Data missing | 53 | 56 |

signature was found highest enriched biological process in LIMA1 high expressing samples (Fig. EV4B). Together, these results suggest that LIMA1 expression promotes the EMT phenotype in HNSCC providing direct evidence that EMT links LIMA1 expression to increase metastatic capability.

Collectively, these results fully support our clinical hypothesis that LIMA1 expression at the time of diagnosis of HNSCC can be used for identification of patients who benefit from cancer surgery, from those who tend to develop metastases and whose cancer treatment requires more extensive adjuvant oncological therapies.

## Discussion

Surgery is the most common cancer treatment modality for HNSCC (Grégoire et al, 2010; Chow, 2020; Bozec et al, 2019; Johnson et al, 2020). However, in current clinical practice, there is no method that could identify those HNSCC patients for whom surgery-only could be a sufficient first-line therapy option.

Therefore, routine HNSCC diagnostics suffers fundamentally from the lack of diagnostic methods to estimate cancer aggressiveness upon primary surgery. Unfortunately, the recent advances in mutation screening, cancer imaging, or patient monitoring, have not provided solutions for this significant clinical problem.

Biomarker studies in general suffer from poor repeatability mostly due to low quality standards by which they have been performed (Ren et al, 2020; Lund-Johansen, 2023). In this study, we have carefully considered the most significant pitfalls of biomarker studies. First, we have robustly validated the antigen specificity of all used antibodies (Fig. EV1) and demonstrate that the results with LIMA1-alpha isoform recognition are consistent with each other when two independent antibodies were used (Figs. 1 and 2). Further, we validate the uniqueness of LIMA1 in predicting poor HNSCC patient survival across seven other biomarkers stained using the same IHC platform. A real strength of our PV-TMA (Mylly et al, 2022; Routila et al, 2021, 2022; Punovuori et al, 2024; Nissi et al, 2025) approach is that we prove that the initial nmHNSCC cohort used for discovery is representative of the average nmHNSCC patient population in the same region. As non-representativeness of the random TMA cohorts used in many biomarker studies is a very important cofounding factor resulting both in false negative and positive results (Ren et al, 2020) we consider our unique approach as one of the important reasons why we could identify such an important novel diagnostic role for LIMA1 in HNSCC. Following that, we validate the IHC results across three other independent patient cohorts out of which two are prospective cohorts collected for this work. Especially the multi-center prospective cohort provided important indication that these results would survive a diagnostic clinical trial setting that we are planning to launch in the next step. Finally, the study complies to REMARK recommendations for reporting prognostic tumor marker. Based on this high-quality biomarker study, we identify low LIMA1-alpha staining as a game-changing diagnostic measure to identify HNSCC patients benefitting from cancer surgery-only therapy and thus avoiding the side effects from adjuvant therapy. The study naturally also has some limitations. First, although LIMA1 low staining has straightforward clinical utility, we did not identify how the prognosis of LIMA1 high patients could be improved. Second, while the number of independent patient cohorts in a multicenter setting was used, we have not yet validated the results in international cohorts. Third, HPV DNA or mRNA level analyses were not available for the patient cohorts in which we performed LIMA1 IHC staining. However, as p16 IHC rather gives a higher number of positive results than molecular detection of HPV (Nissi et al, 2025) and even with p16 staining only 5.9% of patients in our patient cohorts were positive. Therefore, HPV DNA or mRNA detection would not have changed the conclusion of this manuscript regarding the prevalence and correlations of LIMA1 and p16/HPV. Fourth, this study did not address the compatibility of LIMA1 IHC with artificial intelligence-assisted tissue imaging.

Earlier studies have also indicated LIMA1 as a potential prognostic marker in HNSCC, but results from these studies are conflicting, and the studies were done from unselected patient cohorts, and without any focus on the potential role of LIMA1 isoforms (Ma et al, 2022). Thus, the greatest advance of our study is demonstrating the clear prognostic role for LIMA1 in clinically relevant scenario where a clinician has to decide whether to attempt for curative intent surgery knowing that about half of those patients

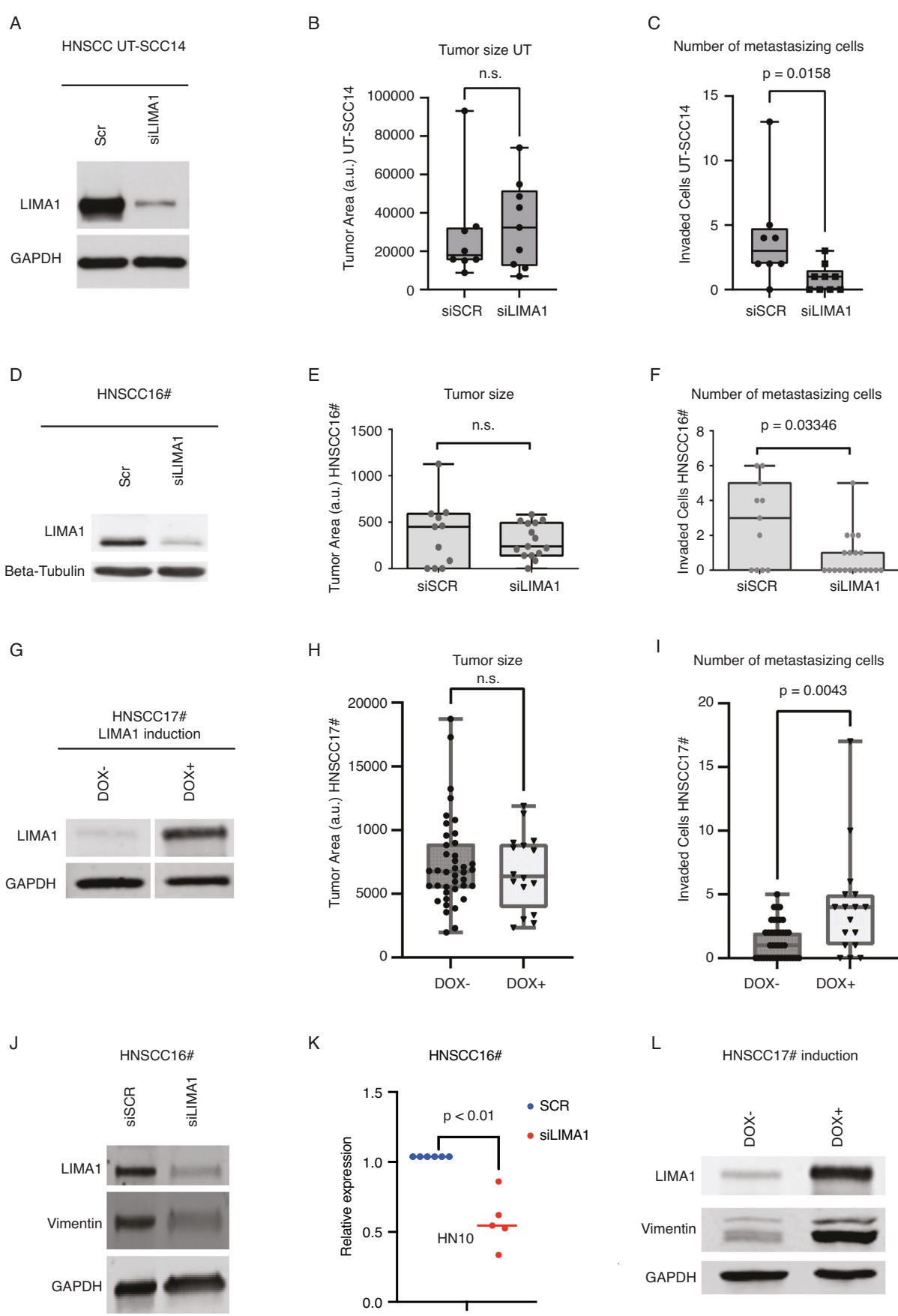

**Figure 4. LIMA1 promotes HNSCC in vivo invasion, metastasis, and EMT.**

(A) Western blot analysis of UT-SCC-14 cells for LIMA1 silencing after siRNA transfection. (B) Measurement of primary tumor size in zebrafish embryo xenograft experiment with UT-SCC-14 cells. siCTRL, $n = 8$ embryos; siLIMA1, $n = 9$ embryos. (C) Quantification of the number of invading cells in the zebrafish embryo xenograft experiment with UT-SCC-14 cells. siCTRL, $n = 8$ embryos; siLIMA1, $n = 9$ embryos. (D) Western blot analysis of HNSCC16# cells for LIMA1 silencing after siRNA transfection. (E) Measurement of primary tumor size in the zebrafish embryo xenograft experiment with HNSCC16# cells. siCTRL, $n = 11$ embryos; siLIMA1, $n = 15$ embryos. (F) Quantification of the number of invading cells in the zebrafish embryo xenograft experiment with HNSCC16# cells. siCTRL, $n = 11$ embryos; siLIMA1, $n = 15$ embryos. (G) Western blot analysis of HNSCC17# cells for LIMA1 overexpression after doxycycline induction. (H) Measurement of primary tumor size in the zebrafish embryo xenograft experiment with HNSCC17# cells. Fluorescent-labeled HNSCC17# cells cultured in the presence or absence of doxycycline were transplanted into zebrafish embryos and embryos cultures with doxycycline to induce LIMA1 expression (DOX + ) or without doxycycline for uninduced controls (DOX + ). DOX−, $n = 40$ embryos; DOX +, $n = 16$ embryos. (I) Quantification of the number of invading cells in the zebrafish embryo xenograft experiment with HNSCC17# cells with or without doxycycline induction of LIMA1 overexpression. DOX−, $n = 40$ embryos; DOX +, $n = 16$ embryos. Non-parametric Mann–Whitney test was used for the statistical analysis of zebrafish experiments. (J) HNSCC16# cells were transfected with scrambled (siSCR) or LIMA1 (siLIMA1) siRNA and analyzed 72 h post-transfection. (K) The relative expression of vimentin represents vimentin level normalized to loading control GAPDH ($n = 5$). Unpaired two-tailed $t$ test with Welch's correction was used for statistical analysis. The exact $P$ values were as indicated. (L) Western blot analysis of LIMA1 and vimentin protein levels in response to doxycycline-induced LIMA1 overexpression in HNSCC17# cells. Cells were treated with 1 μg/ml doxycycline for 2 weeks. Source data are available online for this figure.

will develop later a metastatic disease, or whether to combine surgery with adjuvant therapy causing significant side effects that could have been avoided. Most remarkably, our results from two independent prospective patient cohorts collected for this study, provide real-world evidence that negative LIMA1 staining can identify those nmHNSCC patients that do not die of HNSCC when treated with curative intent surgery. These are revolutionary results in a disease in which there currently is no biomarkers in clinical use that could be used for patient treatment stratification. On the other hand, our results strongly indicate that LIMA1-alpha is the primary LIMA1 isoform driving metastatic HNSCC. These conclusions are supported by a recent study also implicating LIMA1 in the invasion and metastasis in breast cancer, MALT lymphoma and cholangio-sarcoma (Jäntti et al, 2024; Nie et al, 2015; Obulkasim et al, 2024). In addition, recent studies support the view that LIMA1 may influence tumor progression by regulating tumor-infiltrating cells in the tumor microenvironment (TME), suggesting that LIMA1 may be a potential target for immunotherapy (Huang et al, 2022). Our own findings strongly suggest that future studies should focus on isoform-selective LIMA1 detection and targeting that could provide significant help in future cancer diagnostics and anti-metastasis therapies across different cancer types.

In conclusion, our results implicate LIMA1-alpha as a clinical practice-changing biomarker for identifying the HNSCC (1) tumors with high risk of developing into disseminated disease, and (2) patients benefiting from cancer surgery. In clinical use, LIMA1 IHC staining combined with current HNSCC diagnostic routines would constitute a simple and affordable upfront HNSCC diagnostic method for planning of the individualized patient therapy and follow-up strategies based on the tumor aggressiveness.

# Methods

### Reagents and tools table

| Reagent/ resource | Reference or source | Identifier or catalog number |
| --- | --- | --- |
| **Experimental models** | | |
| UT-SCC14 | Auria Biobank | N/A |
| UT-SCC45 | Auria Biobank | N/A |

| Reagent/ resource | Reference or source | Identifier or catalog number |
| --- | --- | --- |
| UT-SCC60B | Auria Biobank | N/A |
| HNSCC16 | This study | N/A |
| HNSCC17 | This study | N/A |
| **Antibodies** | | |
| LIMA1 | Rabbit polyclonal, Atlas Antibodies | HPA023871 |
| | Mouse monoclonal, Santa Cruz Biotechnology | sc-136399 |
| LIMA1- beta | Rabbit polyclonal, custom-made by Biomatik LLC | RB581 |
| CIP2A | Mouse monoclonal, Santa Cruz Biotechnology | sc-80659 |
| OCT4 | Mouse monoclonal, Santa Cruz Biotechnology | sc-5279 |
| NDFIP1 | Rabbit polyclonal, Atlas Antibodies | HPA009682 |
| P16, TP53, MET, EGFR | Ventana staining platform (Roche Diagnostics), clinical pathology laboratory | |
| LIMA1- alpha | Atlas Antibodies, custom-made | RB1201 |
| GAPDH | HyTest Ltd | 5G4-6C5 |
| Vimentin | Cell Signaling Technology | D21H3 |
| Beta-tubulin | Sigma-Aldrich | T7816 |
| Secondary antibodies | Dako | P0447 and P0399 |
| **Chemicals, enzymes, and other reagents** | | |
| DMEM | Sigma | D6429 |
| DMEM | Corning | 10-013-CV |
| FBS | Gibco | 15898937 |
| L- glutamine | Sigma-Aldrich | G7513 |
| Pencillin | | |
| Streptomycin | | |
| Non-essential amino acids | Sigma-Aldrich | M7145 |
| Sodium pyruvate | Gibco | 11360-039 |
| Oligofectamine | Thermo Fisher Scientific | 12252011 |

| Reagent/resource | Reference or source | Identifier or catalog number |
|---|---|---|
| RIPA buffer | In house | N/A |
| Protease inhibitor | Roche | 4693159001 |
| Phosphatase inhibitor | Roche | 4906837001 |
| SDS loading buffer | In house | N/A |
| Precast protein gels | Biorad | 456-1093 and 456-1096 |
| PVDF membranes | Biorad | 1704156 |
| ECL western blotting substrate | Pierce | 32106 |
| PureCol EZ Gel | Sigma | 5074-35 ML |
| 8 µM ThinCert | Greiner bio-one | 665638 |
| LIMA1 (NM_016357.5) | Vector Builder Inc., Chicago, IL, USA | N/A |
| Polybrene | Sigma-Aldrich | TR-1003-50UL |
| CellTracker Green CMFDA | Thermo Fisher Scientific | C7025 |
| **Instruments** | | |
| Nanoject II microinjector | Drummond Scientific | |
| TMA Grand Master | 3D Histech | |
| **Software** | | |
| FIJI | FIJI software | |
| Omero | ImageJ | |
| ImageJ | ImageJ | |
| GraphPad Prism | Graphpad software | version 9 |
| SPSS 28 software | SPSS, IBM | |

## Clinical data

The background HNSCC patient cohort 1 of this study was a previously collected population-validated TMA (tissue microarray) material, covering all new HNSCC patients treated in Southwestern Finland between 2005 and 2010 (Routila et al, 2021, 2022). From this patient material, all 312 patients with non-metastatic HNSCC were selected for the study (Table 1). Metastasis at the time of diagnoses were ruled out by careful clinical investigation, upper panendoscopy, and pertinent imaging (mostly contrast CT), as per national guidelines applied at the time of cohort entry. Overall survival (OS) was defined from end-of-treatment to end-of-follow-up or death. Of these patients, 128 had samples from the tumor resection available in a previously constructed population-validated TMA (Routila et al, 2021). Formalin-fixed, paraffin-embedded (FFPE) tissue samples were acquired from pathology archives through Auria Biobank. TMA blocks with duplicate core biopsies were made using TMA Grand Master (3D

Histech). The study was conducted according to the guidelines of the WMA Declaration of Helsinki and Department of Health and Human Services Belmont Report and approved by the Finnish National Supervisory Authority for Welfare and Health (V/39706/2019) and regional ethics committee of University of Turku (51/1803/2017 and 166/1801/2015).

TCGA patient data, comprising a total of 528 HNSCC patients (HNSCC cohort 2) and 176 surgically treated pancreatic cancer patients (pancreatic cancer cohort), was downloaded from publicly available portals to collect clinical and prognostic data and RNA sequencing data. RNA sequencing results were analyzed using fragments per kilobase of transcript per million fragments mapped (FPKM) values.

The retrospective HNSCC patient cohort 4 containing formalin-fixed paraffin-embedded (FFPE) TMA samples from patients treated for new HNSCC in the Turku University Hospital region between 2011 and 2015 with known TNM staging and survival endpoints (Routila et al, 2021). From this cohort 185 HNSCC patients treated with curative intended surgery as a first-line treatment option were identified and analyzed. In the prospective single-institute HNSCC cohort 3 ($n = 15$) and national multicenter study (HNSCC cohort 5), newly diagnosed HNSCC patients with curative intent surgery were selected for the study. In HNSCC cohort 5, all new early-stage (T1-2N0M0) oral cavity carcinomas were identified and recruited for the study in each of Finland's five university hospitals (Turku, Helsinki, Tampere, Kuopio and Oulu) in accordance with the Medical Ethics Committee of Hospital District of Southwest Finland (100/1801/2017). Patients under 18 years of age, patients unable to give informed consent, and patients with earlier head and neck malignancy were contd. A total of 96 patients met the study criteria, and tissue samples for histology were available from 86 patients.

### Immunohistochemical staining

For immunohistochemistry, previously established and published protocols were followed (Routila et al, 2021, 2022). Following antibodies were used for IHC: LIMA1 (HPA023871 (1:1000 Rabbit polyclonal, Atlas Antibodies)) or (sc-136399 (1:1000 mouse monoclonal, Santa Cruz Biotechnology)), LIMA1-beta (RB581 (1:1000 rabbit polyclonal, custom-made by Biomatik LLC)), CIP2A (sc-80659 (1:25 mouse monoclonal, Santa Cruz Biotechnology)) (Ventelä et al, 2014), OCT4 (sc-5279 (1:20 mouse monoclonal, Santa Cruz Biotechnology)) (Routila et al, 2021; Ventelä et al, 2014), NDFIP1 (HPA009682 (1:1000 rabbit polyclonal, Atlas Antibodies)). P16 (Mylly et al, 2022), TP53 (Routila et al, 2021), MET (Khan et al, 2020), and EGFR (Routila et al, 2021) IHC were stained using Ventana staining platform in the clinical pathology laboratory. Specificities of the used LIMA1 antibodies were confirmed by knockdown of LIMA1 with siRNA followed by western blot analyses and immunofluorescence staining (Fig. EV1). Immunohistochemical staining scorings (negative/low vs. positive) were analyzed by at least two independent investigators, and differences were entered until consensus was reached.

### Cell culture and transfection

All the head and neck cancer cell lines, like UT-SCC-14, UT-SCC-45 and UT-SCC-60 were cultured in DMEM (Sigma) supplemented with 10% heat-inactivated FBS (Gibco), 2 mmol/L L-glutamine,

penicillin (50 units/mL), and streptomycin (50 mg/mL). All cell lines were cultured in a humidified atmosphere of 5% $CO_2$ at 37 °C. Cell lines are regularly subjected to in vitro mycoplasma testing according to laboratory protocols. Small interfering RNA (siRNA) transfections were performed with Oligofectamine™ Transfection Reagent (Thermo Fisher Scientific) following to the manufacturer's protocol. Three days after transfections, cells were harvested for analysis.

## Immunoblotting

Cultured cells or tumor samples were lysed in RIPA buffer (50 mM Tris-HCl pH 7.5, 0.5% DOC, 0.1% SDS, 1% NP-40, and 150 mM NaCl) with protease and phosphatase inhibitors (4693159001 and 4906837001, Roche). The lysate was sonicated, added with 6X SDS loading buffer, boiled and resolved by 4–20% precast protein gels (456-1093 and 456-1096, Biorad). Proteins were transferred to PVDF membranes (1704156, Biorad). Membranes were blocked in 5% Milk-TBS-Tween 20 for 30 min under RT, and then incubated with primary antibodies overnight at 4 °C. Secondary antibodies were incubated in 5% Milk-TBS-Tween 20 for 1 h under RT, and developed by ECL western blotting substrate (32106, Pierce). The following antibodies were used in western blot: LIMA1 antibody (sc-136399, Santa Cruz, HPA023871, Atlas Antibodies, custom-made polyclonal LIMA1-alpha (RB1201) or LIMA1-beta (RB581), GAPDH (5G4-6C5, HyTest Ltd), vimentin (D21H3, Cell Signaling Technology), beta-tubulin (T7816, Sigma-Aldrich). Secondary antibodies were from Dako (P0447 and P0399).

## Inverted invasion assay

Inverted invasion assays were performed as described in a previous study (1). In brief, 200 ml of collagen I (concentration 5 mg/ml; PureCol EZ Gel, Sigma) mixed with 25 mg/ml fibronectin was added to polymerize in inserts (8 µm ThinCert; Greiner bio-one). After warming 1 h at 37 °C, inserts were then inverted, and cells were seeded directly onto the opposite face of the filter. After 4 h of growing, the inverts were inverted again. Transwell inserts were placed in serum-free medium, and medium supplemented with 10% FCS was placed on top of the matrix, providing a chemotactic serum gradient. Migrating cells were fixed 48 h after seeding using 4% PFA for 2 h, permeabilized in 0.5% (vol/vol) Triton-X 100 for 30 min at room temperature. After permeabilization, the inverts were incubated with Alexa Fluor 488 phalloidin overnight at 4 °C. Plugs were then washed three times using PBS and imaged on a confocal microscope (LSM780; Zeiss). Invasion was quantified using the area calculator plugin in ImageJ, measuring the fluorescence intensity of cells invading 45 mm or more and expressing this as a percentage of the fluorescence intensity of all cells within the plug.

## Patient-derived HNSCC samples

Surgical biopsy samples were collected for western blot ex vivo experiments after patient informed consent and in accordance with the local research ethics council permit (Dnro 166/1801/2015). From each patient, a biopsy of tumor tissue and a biopsy of adjacent macroscopically normal tissue from tumor perimeter were collected during scheduled surgery. The samples were snap-frozen

**The paper explained**

**Problem**

Surgery is the most frequent cancer therapy modality. However, there are currently no diagnostic solutions to identify patients who could be stratified to surgery alone.

**Results**

To identify a biomarker that would predict treatment response to cancer surgery, we examined the ability of several biomarkers to detect higher mortality in a retrospective cohort of non-metastatic head and neck squamous cell carcinoma (nmHNSCC). Immunohistochemical staining, which is a routine diagnostic technique used in hospitals worldwide, was chosen as the study method. The best-performing antibody was further validated in independent retrospective population-validated tissue microarray (PV-TMA) and prospective HNSCC cohorts. LIMA1 immunohistochemistry (IHC) with specificity-validated antibodies outperformed all other biomarkers in multivariable survival analyses of patients with nmHNSCC. The prognostic effect was selective to LIMA1-alpha isoform IHC detection in three independent HNSCC cohorts among patients who had received surgical therapy as a first-line treatment option.

**Impact**

Our study provides a novel and cost-effective diagnostic LIMA1 IHC assay for nmHNSCC patient stratification to surgery-only therapy. The application of LIMA1 detection in routine nmHNSCC diagnostics brings a long-needed method to the diagnostics of HNSCC primary cancers.

in liquid nitrogen and stored in a freezer. HNSCC16# and HNSCC17# cancer cells were cultured in DMEM (Corning) supplemented with 5% Tet-approved fetal bovine serum, 2 mM glutamine, 1% non-essential amino acids, 1% sodium pyruvate and 100 U/ml penicillin and streptomycin. LIMA1 [NM_016357.5] lentiviral Tet-inducible gene expression vector systems were purchased from VectorBuilder Inc. (Chicago, IL, USA). Viral transductions were performed HNSCC17# using standard protocol provided by the manufacturer in the presence of 6 µg/ml polybrene (Sigma-Aldrich). For achieving Tet-inducible gene expression, tetracycline was used at 1 µg/ml concentration.

## Zebrafish embryo xenograft experiments

Zebrafish experiments were performed under license (ESAVI/ 31414/2020 issued by Animal Experimentation Board of Regional State Administrative Agency for Southern Finland) and according to European Union Directive 2010/63/EU and essentially as described in more detail earlier (Paatero et al, 2018). The embryos were obtained through natural spawning and incubated at 28.5 °C in E3 medium until xenografted. At 2 days post fertilization, the embryos were xenografted with app. 400–500 tumor cells (UT-SCC-14, HNSCC16# or HNSCC17#) pre-treated with siRNA or doxycycline and labeled with CellTracker Green CMFDA (Thermo Fisher Scientific) were injected into either the yolk or the pericardiac space using Nanoject II microinjector (Drummond Scientific). In doxycycline induction experiments, 5 µg/ml of doxycycline was added to the embryo culture medium, and induction was maintained through the whole experiment. One day after injection, the successfully xenografted embryos were

identified under AxioZoom V16 (Zeiss) fluorescence stereomicroscope and transferred to a 96-well imaging plate. At 4 dpi, the embryos were anaesthetized and imaged using a Nikon Eclipse Ti2 wide-field microscope using ×2 objective and brightfield illumination and GFP fluorescence using filters (excitation 475 nm/28 nm, emission filter GFP sPx 515/30 nm). The images were analyzed manually using FIJI and Omero. Embryos were neither blinded nor formally randomized.

## Statistical analyses

Statistical analyses were carried out with GraphPad Prism (version 9) using non-parametric Mann–Whitney tests. Clinical patient data for each cohort and expression data were entered into SPSS 28 software (SPSS, IBM). For analysis of TMA inclusion bias, logistic regression was used (Burtness et al, 2019). For retrospective patient cohorts, univariate survival was estimated using the Kaplan–Meier method for plotting the survival curves, log-rank method for significance. For multivariable survival analysis, Cox proportional hazards models were constructed. For Cox proportional hazards models, the proportionality of hazards was tested using log-minus-log plotting and plotting Schoenfeld residuals against survival time, when appropriate. Fisher's exact test and Kaplan–Meier survival estimation were used for the prospective cohort.

## Data availability

Individual patient data cannot be shared due to privacy or ethical restrictions. Requests for de-identified and aggregated research data can be sent to the corresponding author.

The source data of this paper are collected in the following database record: biostudies:S-SCDT-10_1038-S44321-025-00266-8.

## Peer review information

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

## Acknowledgements

The authors thank the Auria Biobank for their kind collaboration in preparing the TMA. The authors acknowledge the Turku Bioscience Centre Zebrafish core facility funded by Biocenter Finland. The histological methods were performed by the Histology core facility of the Institute of Biomedicine, University of Turku, Finland. The authors acknowledge Professor Carlos Caldas for informative discussions about biomarker study criteria and Professor Johanna Ivaska for sharing their results related to LIMA1 in breast cancer. This research was funded by the Finnish Medical Foundation, the Finnish ORL-HNS Research Foundation, Jane and Aatos Erkko Foundation (JW and SV), State Research Funding, the Finnish Cultural Foundation, Academy of Finland (Decision number: 354599) and Business Finland Research to Business Funding (Diary number: 1278/31/2021). Research Council of Finland's Flagship InFlames (Decision numbers: 337530 and 357910).

## Author contributions

**Xi Qiao**: Conceptualization; Formal analysis; Investigation. **Johannes Routila**: Conceptualization; Formal analysis; Investigation; Visualization. **Mari Tienhaara**: Investigation. **Heikki Irjala**: Conceptualization; Resources; Supervision; Validation; Investigation; Methodology; Writing—review and editing. **Priyadharshini Parimelazhagan Santhi**: Software; Formal analysis; Validation; Investigation; Visualization; Writing—review and editing. **Teemu Huusko**: Data curation; Software; Formal analysis; Validation; Investigation; Visualization; Methodology; Writing—review and editing. **Linda Nissi**: Data curation; Software; Formal analysis; Supervision; Validation; Investigation; Visualization; Writing—review and editing. **Ilkka Paatero**: Software; Formal analysis; Validation; Investigation; Visualization; Methodology; Writing—original draft. **Noora Lehtinen**: Conceptualization; Software; Investigation; Methodology; Writing—original draft. **Juha Rantala**: Conceptualization; Software; Investigation; Methodology; Writing—original draft. **Toni Viljanen**: Formal analysis; Investigation; Methodology. **Ilmo Leivo**: Formal analysis; Validation; Investigation. **Petri Koivunen**: Validation; Investigation; Methodology; Writing—original draft; Writing—review and editing. **Anna Jouppila-Mättö**: Validation; Investigation. **Rami Taulu**: Validation; Investigation; Methodology; Writing—original draft. **Leif Bäck**: Validation; Investigation; Methodology; Writing—original draft. **Tommy Wilkman**: Validation; Investigation; Methodology; Writing—original draft. **Eeva Haapio**: Validation; Investigation; Methodology; Writing—original draft. **Ilpo Kinnunen**: Validation; Investigation; Methodology; Writing—original draft. **Kari Kurppa**: Conceptualization; Supervision; Validation; Investigation; Methodology; Writing—review and editing. **Jukka Westermarck**: Conceptualization; Data curation; Supervision; Funding acquisition; Project administration; Writing—review and editing. **Sami Ventelä**: Conceptualization; Resources; Data curation; Formal analysis; Supervision; Funding acquisition; Visualization; Project administration; Writing—review and editing.

Source data underlying figure panels in this paper may have individual authorship assigned. Where available, figure panel/source data authorship is listed in the following database record: biostudies:S-SCDT-10_1038-S44321-025-00266-8.

## Disclosure and competing interests statement

The University of Turku has filed a patent covering the diagnostic use of LIMA1 detection on behalf of SV and JW. SV and JW are co-founders and have an ownership interest in Thestra Ltd., developing LIMA1 IHC diagnostics for clinical use.

# Expanded View Figures

**Figure EV1.  Validation of LIMA1 antibodies.** ▶

(**A**) Schematic presentation of LIMA1-alpha and -beta isoforms and the antigen regions used for raising the LIMA1 antibodies used in this study. (**B**) Western blot characterization of LIMA1 expression (HPA023871 antibody) in 14 different HNSCC cancer cell lines and (**C**) in 13 triple-negative breast cancer cell lines. (**D**) Western full blot analysis of LIMA1 antibody (SC-136399) specificity after LIMA1 silencing with six different LIMA1 siRNA transfections. (**E**) Western full blot characterization of custom-made polyclonal LIMA1-beta specific antibody (RB581). UT-SCC-14 and -72 are highly LIMA1-alfa-positive HNSCC cell lines. HN09 HNSCC cancer cell line containing low LIMA1-beta expression were transduced either with LIMA1-alpha (a-induced) or LIMA1-beta (b-induced) lentiviral Tet-inducible gene expression vector. LIMA1 expression and antibody specificity upon Doxycycline treatment were ensured by western blot with LIMA1-beta specific antibody (RB581). (**F, G**) Western blot and immunofluorescence analyzes of HPA023871 and SC-136399 antibodies after LIMA1 siRNA silencing of patient-derived HNSCC cell lines. (**H**) Specificity in immunofluorescence of LIMA1-beta specific antibody (RB581) was ensured by conducting RB581 IF staining in HNSCC cell lines containing both LIMA1-alpha and -beta expression (MISB10) and with the cell line lacking LIMA1-beta expression (UT-SCC-14). Scale bars indicated were 10 µm.

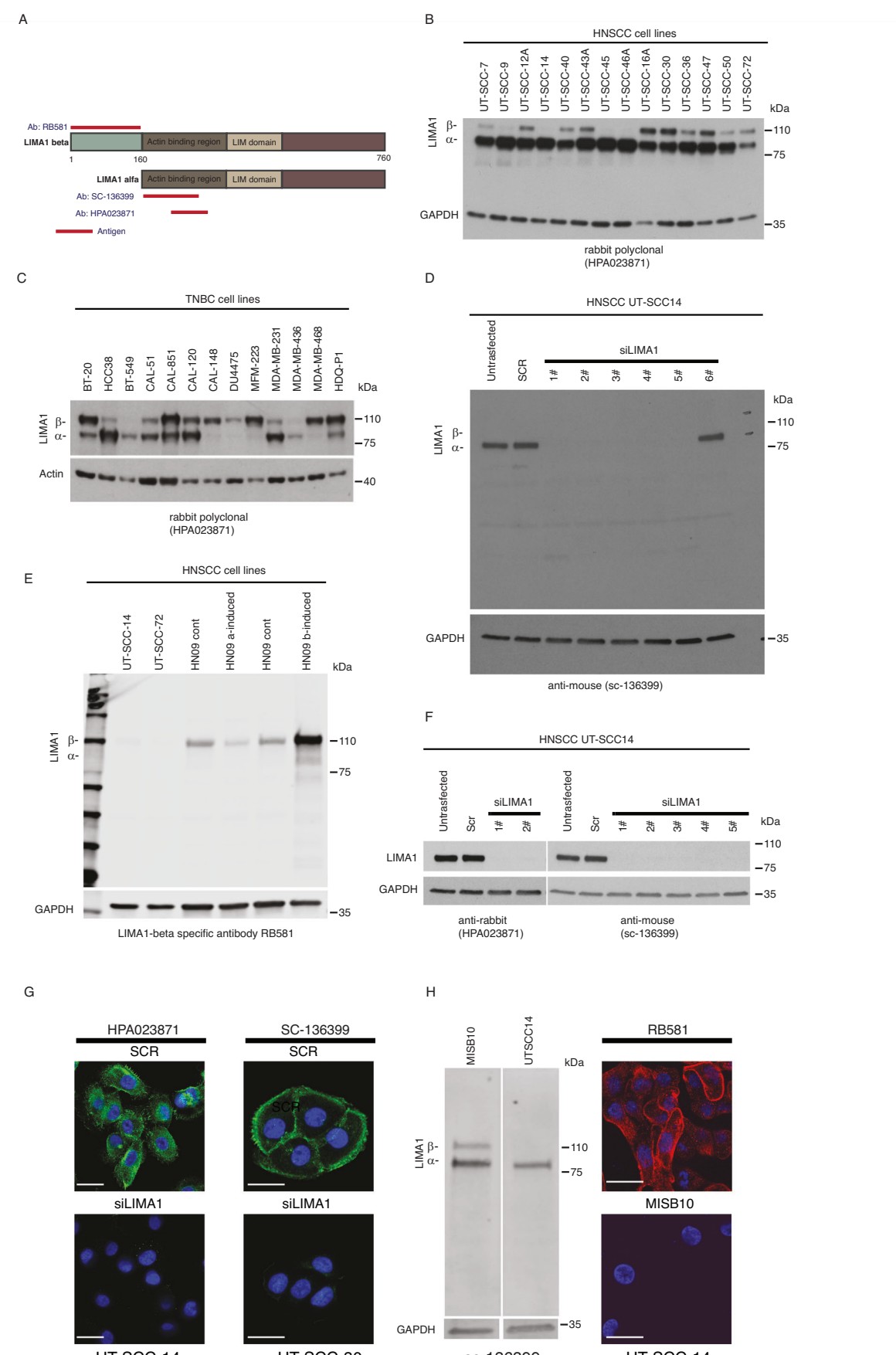

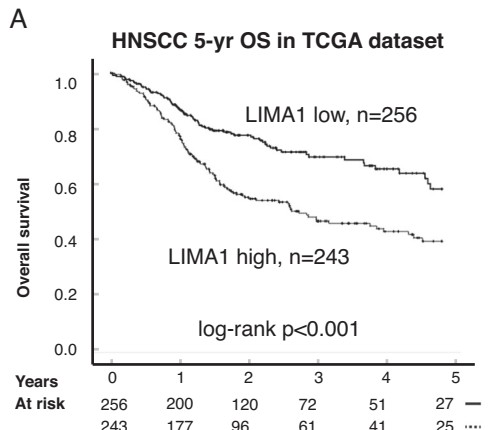

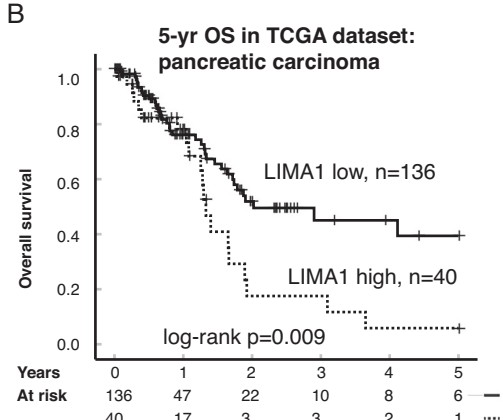

**Figure EV2.   Prognostic role for LIMA1 in TCGA HNSCC and pancreatic cancer data.**

(A, B) The prognostic effect of LIMA1 was confirmed in TCGA HNSCC ($P < 0.001$) and operatively treated pancreatic cancer ($P = 0.009$) datasets.

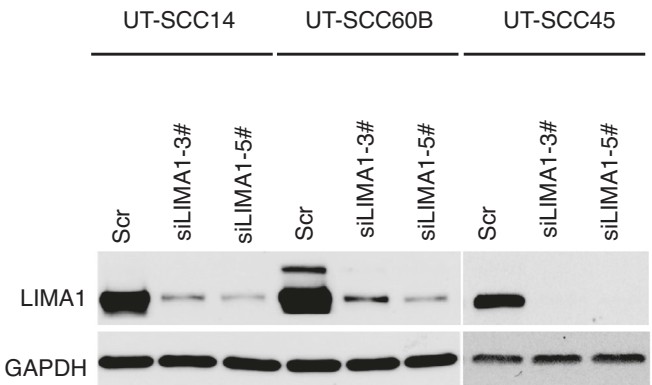

**Figure EV3.  Validation of LIMA1 protein inhibition by siRNA.**

Depletion of LIMA1 in three different patient-derived HNSCC cell lines (UT-SCC14, UT-SCC60B and UT-SCC45).

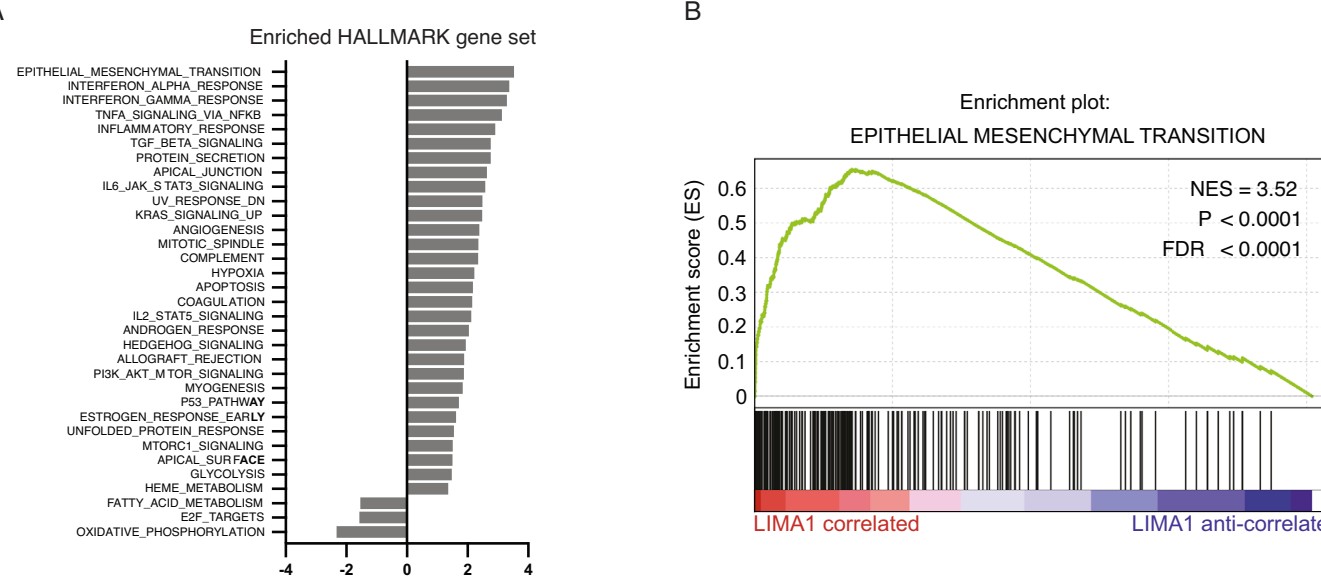

**Figure EV4.  LIMA1 promotes epithelial mesenchymal transition.**

(**A**) Gene set enrichment analysis (GSEA) for genes co-expressing with LIMA1 in TCGA HNSCC data set. The normalized enrichment scores (NES) for HALLMARK gene sets with FDR < 0.05 are shown. (**B**) Enrichment plot for HALLMARK EMT gene set (*P* < 0.0001) from GSEA analysis. Default statistics of GSEA was used (Subramanian et al, 2005).

