## [Peer Review File · EMBO Molecular Medicine]

LIMA1-alpha staining predicts curative intent surgery response in HPV negative head and neck cancer

Xi Qiao, Johannes Routila, Mari Tienhaara, Heikki Irjala, Priyadharshini Santhi, Teemu Huusko, Linda Nissi, Ilkka Paatero, Noora Lehtinen, Juha Rantala, Toni Viljanen, Ilmo Leivo, Petri Koivunen, Anna Jouppila-Mättö, Rami Taulu, Leif Bäck, Tommy Wilkman, Eeva Haapio, Ilpo Kinnunen, Kari Kurppa, Jukka Westermarck, and Sami Ventelä

Corresponding author(s): Jukka Westermarck (jukwes@utu.fi) , Sami Ventelä (satuve@utu.fi)

Review Timeline:

Submission Date:	21st Oct 24
Editorial Decision:	11th Nov 24
Revision Received:	10th Mar 25
Editorial Decision:	21st Mar 25
Revision Received:	8th Jun 25
Accepted:	16th Jun 25

Editor: Zeljko Durdevic

Transaction Report:

11th Nov 2024

Dear Prof. Westermarck,

Thank you for the submission of your manuscript to EMBO Molecular Medicine. We have now received feedback from the two reviewers who agreed to evaluate your manuscript. Both referees recognize potential interest of the study but also raise serious and partially overlapping concerns, particularly regarding the lack of separation of HPV positive and HPV negative HNSCCs.

Taking referee concerns in consideration it is clear that publication of the paper cannot be considered at this stage. After an editorial discussion and our cross-commenting session we agreed that further consideration of the manuscript will depend on the analysis outcome after separation of HPV positive and HPV negative patients. Therefore, we would like to invite major revision that addresses reviewers' concerns in full with the focus on the LIMA1 predictive power depending on the HPV status of HNSCC patients and performing multivariate comparisons. If you would like to discuss further the points raised by the referees, I am available to do so via email or video. Let me know if you are interested in this option.

Further consideration of the revised manuscript will entail a second round of review. EMBO Molecular Medicine encourages a single round of revision only and therefore, acceptance or rejection of the manuscript will depend on the completeness of your responses included in the next, final version of the manuscript. For this reason, and to save you from any frustrations in the end, I would strongly advise against returning an incomplete revision.

We would welcome the submission of a revised version within three months for further consideration. Please let us know if you require longer to complete the revision.

I look forward to receiving your revised manuscript.

Yours sincerely,

Zeljko Durdevic

We require:

- 1) A .docx formatted version of the manuscript text (including legends for main figures, EV figures and tables). Please make sure that the changes are highlighted to be clearly visible.
- 2) Individual production quality figure files as .eps, .tif, .jpg (one file per figure). For guidance, download the 'Figure Guide PDF': (<https://www.embopress.org/page/journal/17574684/authorguide#figureformat>).
- 3) A .docx formatted letter INCLUDING the reviewers' reports and your detailed point-by-point responses to their comments. As part of the EMBO Press transparent editorial process, the point-by-point response is part of the Review Process File (RPF), which will be published alongside your paper.
- 4) A complete author checklist, which you can download from our author guidelines (<https://www.embopress.org/page/journal/17574684/authorguide#submissionofrevisions>). Please insert information in the checklist that is also reflected in the manuscript. The completed author checklist will also be part of the RPF.

6) It is mandatory to include a 'Data Availability' section after the Materials and Methods. Before submitting your revision, primary datasets produced in this study need to be deposited in an appropriate public database, and the accession numbers and database listed under 'Data Availability'. Please remember to provide a reviewer password if the datasets are not yet public (see <https://www.embopress.org/page/journal/17574684/authorguide#dataavailability>).

12) Author contributions: You will be asked to provide CRediT (Contributor Role Taxonomy) terms in the submission system. These replace a narrative author contribution section in the manuscript.

13) A Conflict of Interest statement should be provided in the main text.

14) Every published paper now includes a 'Synopsis' to further enhance discoverability. Synopses are displayed on the journal webpage and are freely accessible to all readers. They include a short stand first (maximum of 300 characters, including space) as well as 2-5 one-sentences bullet points that summarizes the paper. Please write the bullet points to summarize the key NEW findings. They should be designed to be complementary to the abstract - i.e. not repeat the same text. We encourage inclusion of key acronyms and quantitative information (maximum of 30 words / bullet point). Please use the passive voice. Please attach these in a separate file or send them by email, we will incorporate them accordingly.

15) Include a Reagents and Tools Table as part of the Methods section, which can be downloaded from our author guidelines (<https://www.embopress.org/page/journal/17574684/authorguide#structuredmethods>)

***** Reviewer's comments *****

Referee #1 (Comments on Novelty/Model System for Author):

This manuscript is right in an area that I am quite familiar with. As presented, the utility of a simple standard IHC test for LIMA1 sounds like it has the potential to inform patients with small non-invasive head and neck cancers as to the likelihood of cure outcomes from a surgery only approach. Note that this is not entirely novel as several other reports have made this observation (PMID: 35837368;37181789; 34069237). A role in EMT in HNSCC has also been established in PMID: 35837368, making the mechanistic part of the current submission less novel as well. Granted, the existing papers cited above do not validate their results in other cohorts beyond the TCGA except for PMID: 34069237.

I am somewhat concerned regarding some mistakes/inconsistencies in describing the initial cohort related to p16 status (a surrogate marker for HPV+ HNSCC disease, which has better outcomes than HPV- HNSCC disease) might have made them eliminate the possibility that high LIMA1 is just a feature of HPV- HNSCC, while low LIMA1 is a feature of HPV+ HNSCC. This should be resolved in the multivariate analysis, but the full range of multivariate comparisons in any of the validation cohorts are not included in the manuscript.

Given that HPV- and HPV+ HNSCC have distinct etiology, pathology and patient outcomes, I think it is absolutely essential to establish whether LIMA1 expression is influenced by HPV status. If LIMA1 is really just related to HPV status, the manuscript is uninteresting clinically. I suspect that if LIMA1 has any value clinically, it might be restricted to HPV- HNSCC. This could still be interesting.

Referee #1 (Remarks for Author):

Qiao report the possibility that high LIMA1 protein expression as detected by standard IHC can predict poor patient survival in non-metastatic HNSCC. This relationship is identified in a primary cohort of 312 patients from Finland with 128 samples available for staining in a TMA. Follow up work in several validation cohorts including mRNA expression (not protein) from the TCGA HNSC and several other Finnish HNSCC cohorts analyzed by IHC, including a prospective multicenter cohort that is still accruing. The key LIMA1 isoform seems to be the smaller alpha protein. Some basic science experiments are presented to support the hypothesis that LIMA1 impacts EMT related gene express in cell line models and invasion in wound-healing assays. Metastasis assays in zebrafish also support a role for LIMA1 in EMT.

Importance: As noted by the authors, prognostic markers for HNSCC are definitely needed.

Comments:

1) there seems to be some discrepancy in how the initial cohort is described. Supplementary Table states that 103 patients are p16 negative, yet panel I on Supplementary Figure 2 shows 103 patients as p16 positive. This needs to be sorted out.

2) In HNSCC, HPV+ and HPV- disease have distinct etiology, pathology, immune landscape and patient outcomes as well as subsite prevalence. This probably needs to be incorporated into the manuscript. Despite the authors initial analysis showing that LIMA1 expression is independent of p16 status (a surrogate marker of HPV status), I think it is absolutely essential to establish that LIMA1 expression is independent of HPV status in ALL cohorts presented. This is really important as existing work has shown that HPV+ samples in the TCGA HNSC cohort exhibit low LIMA1 expression and that HPV+ HNSCC patients also exhibit improved prognosis (PMID: 35837368). This is also visible using THInCR webtools (PMID: 35950764, <https://mymryklab.ca/main-home/>), which also shows that OS for HNSC HPV(-) High vs. Low is significantly different ($p=0.0302$) but not for HPV+ HNSC. The initial cohort multivariable analysis significantly correlates survival with oropharynx subsite and

neck dissection, frequently associated with HPV+ HNSCC. I am very concerned that because HPV- HNSCC express higher LIMA1 levels and exhibit worse survival compared to HPV+ HNSCC, this is the reason why LIMA1 works to predict survival.

3) p16 status is not a perfect substitute for HPV status. In this day in age, molecular testing for HPV status data should be available. It certainly is for the TCGA cohort at least. This should be considered directly in the manuscript. Analysis should be done independently for HPV+ and HPV- disease. This may still be a useful marker even if it is only useful in HPV- HNSCC.

4) exon level mRNA values are available for the TCGA HNSC cohort. the authors should be able to determine which LIMA1 species are expressed based on those reads. See PMID: 29069809 for an example related to alternative splicing of the CDKN2A and B locus.

5) on line 341, is the survival difference reported statistically significant?

6) on table 3, n is listed yet only 86 are reported. this would be clearer if n was just reported as 86 analyzed. Subsite should be listed here and HPV status (or at least p16)

7) are all the HNSCC cell lines HPV negative?

8) EMT signatures could be readily calculated for the TCGA HNSCC samples and correlated with LIMA1 expression or even just correlate individual EMT genes with LIMA1 expression.

9) Note that there are huge immunological differences in HNSCC that correlate with LIMA1 expression based on THInCR analysis and similarly reported in PMID: 37181789. This never made it into the intro or discussion.

10) for figure 3d, is this all isoforms of LIMA1 or just the alpha isoform? Not labelled clearly like figure 3F.

11) The work is not entirely novel and is more confirmatory. Several other reports have made similar observations related to LIMA1 (PMID: 35837368;37181789; 34069237). A role FOR LIMA1 in EMT in HNSCC has also been established in PMID: 35837368, making the mechanistic part of the current submission less novel as well. Granted, the existing papers cited above do not validate their results in other cohorts beyond the TCGA except for PMID: 34069237.

Referee #2 (Comments on Novelty/Model System for Author):

HNSCC is an aggressive disease for which treatment can result in dramatic changes to a patients lifestyle. Any biomarker that can lead to de-escalation of treatment would be a significant advance. The authors make a very appealing case of that in this manuscript. They include both retrospective and prospective patient cohorts, they validate commercially available antibodies that could be used in the clinical setting and their survival analyses is convincing.

Referee #2 (Remarks for Author):

The authors propose using the detection of LIMA1 in patient samples as an indicator of good prognosis (no or low expression) and reduced treatment with the expectation of a higher survival rate. Versus patients with detectable high expression of LIMA1 as a predictor of poorer outcomes or the development of metastases.

It is a simple message high LIMA1 is bad and low LIMA1 is good.

There are several very appealing and convincing aspects to this manuscript. The authors employ multiple patient cohorts, both prospective and retrospective to test their hypothesis. They confirm the specificity of two commercial antibodies and develop their own to identify between two isoforms of LIMA1. Their IHC based survival analyses seems very convincing.

I have several issues many revolving around the lack of separation of HPV positive HNSCC and HPV negative HNSCC. These are distinct diseases and should be examined separately.

Issues

My major concern is the lack of identification of the role of HPV positive HNSCC versus HPV negative HNSCC. There is much in

this manuscript that suggests that the authors are only concerned with the HPV negative disease. However, nowhere do they make this clear. It is critical that this be addressed right at the beginning, including the manuscript title. Otherwise it is very confusing. For example, in the introduction the authors state that " approx. 90% of head and neck cancers are squamous cell carcinomas which have a relatively high, approx. 50% five year mortality rate". This is true, but only for HPV negative HNSCC. The survival rate of HPV positive HNSCC is closer to 80-85%.

The title is needs to be revised. I would prefer something like "LIMA1 immunohistochemistry predicts curative intent surgery response in HPV negative head and neck cancer".

The first cohort of this study is a TMA comprised of patient material collected between 2005-2010 in Finland. Perhaps in Finland, as in North America the presence of HPV positive HNSCC was relatively low during those years. However, development of HPV positive HNSCC is growing at epidemic proportions around the world. I am not sure that the TMA is useful other than as an indicator of LIMA levels in HPVnegative disease.

The HNSCC cohort 2 that the authors employ is the head and neck TCGA database (n=528) however, there are 40 normal samples in the collection and 72 (or73) HPV positive samples. This is never mentioned or addressed, and this could be a confounding factor in their analyses. A re-analysis of the data is necessary with a separation and identification of HPV negative patients and HPV positive patients.

The authors state that from their original cohort of 476 patients, they identified 312 cases that did not have any signs of metastasis based on, I assume, clinical reports (which is not stated but should be). "97 patients of this cohort were treated with curative intended surgery". 32 patients died during the 5 years of follow up (these don't add up to the 128 patients available in the cohort (line 199)). did these 32 receive chemo-radiation if they were surgery alone? And did the screen of the TMA from which this data was derived support this separation? Meaning did the 97 that had surgery alone and were still alive at 5 year FU have low or undetectable LIMA1 and the 32 that died have high LIMA1? If this was stated, then I missed it and apologize. Otherwise it should be examined.

Line 261-162 "...prognostic role of LIMA1 could be extended to other cancers types..." which would be useful but the only other cancer type the authors examined was pancreatic cancer. I think that a more types should be examined before I accept that LIMA1 would be widely used.

The data in Figure 2B, C is intriguing; however, the cohort is too small for any general claims.

I applaud the authors efforts and look forward to seeing this work published.

LIMA1-alpha staining predicts curative intent surgery response in HPV negative head and neck cancer

Qiao et al.,

Response to reviewers

We were very delighted to notice that both reviewers clearly appreciated the direct translational importance of our results, and for their very constructive comments how to improve the work even further. We have now addressed all the comments either by adding new data or by modifying the manuscript text (incl. the title) and tables. Based on these revisions the work is even better focused to the clinical scenario where the results can be translated to clinical HNSCC diagnostics. We thank the reviewers for their important efforts and hope that the work is now suitable to be published at *Embo Molecular Medicine*.

***** Reviewer's comments *****

Referee #1 (Comments on Novelty/Model System for Author):

This manuscript is right in an area that I am quite familiar with. As presented, the utility of a simple standard IHC test for LIMA1 sounds like it has the potential to inform patients with small non-invasive head and neck cancers as to the likelihood of cure outcomes from a surgery only approach. Note that this is not entirely novel as several other reports have made this observation (PMID: 35837368;37181789; 34069237). A role in EMT in HNSCC has also been established in PMID: 35837368, making the mechanistic part of the current submission less novel as well. Granted, the existing papers cited above do not validate their results in other cohorts beyond the TCGA except for PMID: 34069237.

Our response: *We are very grateful for the comments from the Referee regarding the literature references, which perfectly support our view that detection of LIMA1 in cancer has great potential for diagnosing HNSCC aggressiveness. However, none of the earlier studies had focused on the defined question whether LIMA1 could function as clinically applicable HNSCC biomarker in curative intent surgery only scenario. Further, the biggest challenges in the LIMA1 literature have the shortcomings in ensuring the specificity of the LIMA1 antibodies, the lack of understanding of LIMA1 isoforms, and especially lack of evidence from prospective patient cohorts mirroring the real-life diagnostic use of LIMA1 antibodies. We have tackled all these challenges in the current work and therefore we strongly feel that the current revised manuscript, addressing also the other reviewer's concerns, provides a significant advance towards using LIMA1 IHC detection for clinical HNSCC diagnostics. We have also strengthened the connection between high LIMA expression and EMT.*

I am somewhat concerned regarding some mistakes/inconsistencies in describing the initial cohort related to p16 status (a surrogate marker for HPV+ HNSCC disease, which has better outcomes than HPV- HNSCC disease) might have made them eliminate the possibility that high LIMA1 is just a feature of HPV- HNSCC, while low LIMA1 is a feature of HPV+ HNSCC. This should be resolved in the multivariate analysis, but the full range of

multivariate comparisons in any of the validation cohorts are not included in the manuscript.

Given that HPV- and HPV+ HNSCC have distinct etiology, pathology and patient outcomes, I think it is absolutely essential to establish whether LIMA1 expression is influenced by HPV status. If LIMA1 is really just related to HPV status, the manuscript is uninteresting clinically. I suspect that if LIMA1 has any value clinically, it might be restricted to HPV- HNSCC. This could still be interesting.

Our response: We are very pleased with the referee's remark about the p16/HPV analysis in our work. This aspect was not particularly emphasized in the manuscript because most patients in our cohorts were from other tumor sites than Oropharynx that is the site from where most of the HPV positive cancers are derived (see Table 1). To clarify this, we have added a new table to the Appendix data (Table EV3), in which we have added p16 staining data from each of our study cohorts. This new table is also attached below. It is noteworthy that in our entire dataset, only 25/423 (5.9%) patients had p16 positive HNSCC tumor. Therefore, the low proportion of p16 positive samples did not allow for statistical analyses to be performed, where LIMA1 expression would have been examined separately in p16 positive or negative HNSCC samples. Furthermore, when we compared LIMA1 prevalence in our studied cohorts (summary table below), we see that LIMA1 Low and High is distributed much more evenly across cohorts than p16, also indicating that LIMA1 expression is not related to HPV status. In order to clarify this important aspect to the reader, we have now modified the title of the paper to indicate that the results are derived by large from HPV negative HNSCC and have included the p16/HPV positivity information to all data tables and text where applicable.

	p16 positive		p16 negative		p16 data missing		n
	n	%	n	%	n	%	
Cohort 1	13	10.2	103	80.4	12	9.4	128
Cohort 3	0	-	7	46.7	8	53.3	15
Cohort 4	8	4.3	175	94.6	2	1.1	185
Cohort 5	4	4.2	38	40	53	55.8	95
In total/average	25	5.9	323	76.3	75	17.7	423
	LIMA1 Low		LIMA1 High		LIMA1 data missing		n
	n	%	n	%	n	%	
Cohort 1	63	49.2	63	49.2	2	1.6	128
Cohort 3	5	33.3	10	66.7	0	0	15
Cohort 4	116	62.7	68	36.8	1	0.5	185
Cohort 5	44	46.3	42	44.2	9	9.5	95
In total/average	228	53.9	183	43.3	12	2.8	423

Referee #1 (Remarks for Author):

Qiao report the possibility that high LIMA1 protein expression as detected by standard IHC can predict poor patient survival in non-metastatic HNSCC. This relationship is identified in a primary cohort of 312 patients from Finland with 128 samples available for staining in a TMA. Follow up work in several validation cohorts including mRNA expression (not protein) from the TCGA HNSC and several other Finnish HNSCC cohorts analyzed by IHC, including a prospective multicenter cohort that is still accruing. The key LIMA1 isoform seems to be the smaller alpha protein. Some basic science experiments are presented to support the hypothesis that LIMA1 impacts EMT related gene express in cell line models and invasion in wound-healing assays. Metastasis assays in zebrafish also support a role for LIMA1 in EMT.

Importance: As noted by the authors, prognostic markers for HNSCC are definitely needed.

Comments:

1) there seems to be some discrepancy in how the initial cohort is described. Supplementary Table states that **103 patients are p16 negative, yet panel I on Supplementary Figure 2 shows 103 patients as p16 positive**. This needs to be sorted out.

***Our response:** Thank you for this observation. The p16 negative and positive patient numbers were written incorrectly in Supplementary Figure 2 I. This unfortunate error has now been corrected in the revised manuscript (Appendix Figure S1).*

2) In HNSCC, HPV+ and HPV- disease have distinct etiology, pathology, immune landscape and patient outcomes as well as subsite prevalence. This probably needs to be incorporated into the manuscript.

***Our response:** We have now addressed this in the introduction (In. 99-102)*

Despite the authors initial analysis showing that LIMA1 expression is independent of p16 status (a surrogate marker of HPV status), **I think it is absolutely essential to establish that LIMA1 expression is independent of HPV status in ALL cohorts presented**. This is really important as existing work has shown that HPV+ samples in the TCGA HNSC cohort exhibit low LIMA1 expression and that HPV+ HNSCC patients also exhibit improved prognosis (PMID: 35837368). This is also visible using THInCR webtools (PMID: 35950764, <https://mymryklab.ca/main-home/>), which also shows that OS for HNSC HPV(-) High vs. Low is significantly different ($p=0.0302$) but not for HPV+ HNSC. The initial cohort multivariable analysis significantly correlates survival with oropharynx subsite and neck dissection, frequently associated with HPV+ HNSCC. I am very concerned that because HPV- HNSCC express higher LIMA1 levels and exhibit worse survival compared to HPV+ HNSCC, this is the reason why LIMA1 works to predict survival.

***Our response:** We are very grateful to Referee for her/his very precise observations regarding the co-expression of LIMA1 and p16 / HPV. We have now collected p16 expression data from all our HNSCC cohorts and added a summary table to the Appendix data (Supplementary Table EV3). As already explained in our response above, the great majority of patients (398/423) in our HNSCC cohorts are HPV negative, and thus the study basically focuses only on HPV negative disease. Therefore, the prognostic role for LIMA1 in these cohorts cannot be explained by their HPV status. To clarify this important aspect*

to the reader, we have now modified the title of the paper to indicate that the results are derived by large from HPV negative HNSCC and have included the p16/HPV positivity information to all data tables and text where applicable.

Additional note for Reviewer use only: *In our follow-up study in progress we have focused on LIMA1-alpha role in the HPV positive HNSCC site (OPSCC). In multivariate analyses LIMA1 was significantly better prognosticator compared to p16 IHC or HPV cish in OPSCC. These studies in combination thus demonstrate that LIMA1 serves as a prognostic biomarker in both HPV-negative and HPV-positive HNSCCs. The results of the latter study will be submitted for publication by the summer 2025. Once published, these two studies will together firmly establish LIMA1 as prognostic biomarker across most HNSCC subtypes.*

3) p16 status is not a perfect substitute for HPV status. In this day in age, molecular testing for HPV status data should be available. It certainly is for the TCGA cohort at least. This should be considered directly in the manuscript. Analysis should be done independently for HPV+ and HPV- disease. This may still be a useful marker even if it is only useful in HPV- HNSCC.

Our response: *We absolutely agree with the Referee regarding the limitations of p16 immunohistochemistry in detecting HPV positivity and the need for molecular testing for HPV to improve the deficiency of p16 IHC staining. Unfortunately, HPV DNA or mRNA level analyses were not available for the patient cohorts in which we performed LIMA1 IHC staining. On the other hand, it is known that p16 IHC is more sensitive in giving a positive result than methods based on HPV RNA or DNA detection. Overall, only 5.9% of patients in our patient cohorts were p16 IHC positive, so using HPV DNA or mRNA detection would likely further reduce the number of HPV positive samples in our cohorts. Therefore, HPV DNA or mRNA detections would not change the conclusion of this manuscript regarding the prevalence and correlations of LIMA1 and p16/HPV. We have now mentioned this aspect as one of the limitations of the study (Ln. 631).*

Furthermore, we are very grateful for the Referee's suggestion to look at the correlations between LIMA1 and HPV expression in TCGA data. As a result, mRNA expression level of LIMA1 was significantly higher in HPV negative samples in the TCGA HNSCC data set when mRNA expressions were compared to HPV positive and normal (figure below). This result further supports our view that high expression of LIMA1 is associated with more aggressive HNSCC. This data has also now been added to Appendix Data.

Comparison Group	p-Value	q-Value	Significance
HNSC - HPV(+) vs. HPV(-)	1.15e-1	3.85e-14	YES
HNSC - HPV(+) vs. Normal,	0.93	0.24	NO
HNSC - HPV(-) vs. Normal,	1.91e-10	3.48e-10	YES

4) **exon level mRNA** values are available for the **TCGA HNSC cohort**. the authors should be able to determine which LIMA1 species are expressed based on those reads. See PMID: 29069809 for an example related to alternative splicing of the CDKN2A and B locus.

Our response: We are very happy with the Referee's suggestion to also look at exon level mRNA values of LIMA1 and HPV from TCGA data. Exon level mRNA expression of LIMA1 were analyzed by using <https://xenabrowser.net/>. Exons 1-3 (first three exons from the left) which are specific for LIMA1 β isoform had no significant association to HPV status when HPV ISH or P16 detection were used. However, HPV negative samples had significant correlation with LIMA1-alpha specific exons starting from exon 4, which clearly indicates that the LIMA1-alpha isoform is connected to poor prognosis and HPV negative status in HNSCC.

*These results strongly support our conclusions that LIMA1-**alpha** serves as a prognostic marker in HPV negative HNSCC. These results have also now been added to revised manuscript (Appendix Figure S3).*

5) on line 341, is the survival difference reported statistically significant?

Our response: *This point has been clarified in the revised manuscript and the statistical significance of the results have been added to the text (ln. 409-412).*

6) on table 3, n is listed yet only 86 are reported. this would be clearer if n was just reported as 86 analyzed. Subsite should be listed here and HPV status (or at least p16)

Our response: *The requested changes have been made to the Table 3.*

7) are all the HNSCC cell lines HPV negative?

Our response: *This question, to what extent HNSCC cell lines represent HNSCC patient tumors, including HPV status, is very interesting and has been studied in several of our projects and publications. To compare and profile HNSCC cell lines, we have, among other things, commissioned multi-cell line blocks (so-called CMA = Cell MicroArrays;*

PMID: 34320941) from different HNSCC cell lines, in which separate HNSCC cell lines in the same section can be simultaneously stained with the same antibody. In the images below, HNSCC cell lines in the left section (each spot corresponds to one HNSCC cell line) have been stained with p16 antibody (CINtec® p16 Histology) and the right section has been stained with LIMA1 antibody (sc-136399). As can be seen from the figures, about 50% of HNSCC cell lines are p16 negative, while all HNSCC cell lines are LIMA1 positive. These results further support our view that LIMA1 expression is not dependent on the HPV/p16 status of HNSCC.

8) EMT signatures could be readily calculated for the TCGA HNSCC samples and correlated with LIMA1 expression or even just correlate individual EMT genes with LIMA1 expression.

Our response: We are very grateful for this Referee's suggestion and performed the requested analyses from the HNSCC TCGA data. Based on the analyses, very strong correlation between EMT and LIMA1 expression existed. These results are included in the revised publication (Ln. 546, Figure EV4).

9) Note that there are huge immunological differences in HNSCC that correlate with LIMA1 expression based on THInCR analysis and similarly reported in PMID: 37181789. This never made it into the intro or discussion.

Our response: *We are very grateful to the Referee for this report. In our revised manuscript, we have added and referenced this work in the Discussion section (In 655).*

10) for figure 3d, is this all isoforms of LIMA1 or just the alpha isoform? Not labelled clearly like figure 3F.

Our response: *This point has now been corrected in the revised manuscript.*

11) The work is not entirely novel and is more confirmatory. Several other reports have made similar observations related to LIMA1 (PMID: 35837368;37181789; 34069237). A role FOR LIMA1 in EMT in HNSCC has also been established in PMID: 35837368, making the mechanistic part of the current submission less novel as well. Granted, the existing papers cited above do not validate their results in other cohorts beyond the TCGA except for PMID: 34069237.

Our response: *We are very grateful for all the comments provided by the Referee #1, which helped us to make our publication much clearer and scientifically stronger. We are especially grateful for all the comments and suggestions related to the clarification of the expression profiles between LIMA1 and HPV. The absolute strength and novelty of our work is elucidation of the role of LIMA1 isoforms in HNSCC prognostication, as well as validation of the results across four different HNSCC cohorts by IHC. Out of these, two were **first ever performed prospective studies that produced real-world evidence for usefulness of LIMA1 detection in future clinical HNSCC diagnostics**. The most striking finding repeated across both prospective studies was that **none of the LIMA1alpha low patients died of HNSCC during follow-up**. Yet another very important advance from our work related to clinical translation of the results is that we have performed the analysis from patient cohorts with curative intended surgery population, giving clear clinical guidelines how to use the approach in clinical practice. Thereby, our results collectively show that if introduced to clinical practice, it is possible to achieve such diagnostic resolution between LIMA1 positive and negative patients that this could establish the first biomarker-based approach for therapy guiding in HNSCC. This is a*

*significant advance from the earlier studies in which statistical significance in survival of HNSCC patients could be demonstrated by LIMA1 staining, but as the survival difference between populations was so marginal that clinicians treating HNSCC patients would hardly be convinced to use LIMA1 IHC as clinical diagnostic approach. Thereby our work do support the other findings but is unique in that sense that we establish **LIMA1 alfa IHC detection as a simple and cost-effective diagnostic method for HNSCC treatment guidance for clinical practice**. This is also why we think Embo Molecular Medicine is a very suitable forum to publish these results with direct translational relevance and clinical applicability.*

Referee #2 (Comments on Novelty/Model System for Author):

HNSCC is an aggressive disease for which treatment can result in dramatic changes to a patient's lifestyle. Any biomarker that can lead to de-escalation of treatment would be a significant advance. The authors make a very appealing case of that in this manuscript. They include both retrospective and prospective patient cohorts, they validate commercially available antibodies that could be used in the clinical setting and their survival analyses is convincing.

Our response: *We are very grateful for these positive comments from the Referee.*

Referee #2 (Remarks for Author):

The authors propose using the detection of LIMA1 in patient samples as an indicator of good prognosis (no or low expression) and reduced treatment with the expectation of a higher survival rate. Versus patients with detectable high expression of LIMA1 as a predictor of poorer outcomes or the development of metastases.

It is a simple message high LIMA1 is bad and low LIMA1 is good.

There are several very appealing and convincing aspects to this manuscript. The authors employ multiple patient cohorts, both prospective and retrospective to test their hypothesis. They confirm the specificity of two commercial antibodies and develop their own to identify between two isoforms of LIMA1. Their IHC based survival analyses seems very convincing.

I have several issues many revolving around the lack of separation of HPV positive HNSCC and HPV negative HNSCC. These are distinct diseases and should be examined separately.

Issues

My major concern is the lack of identification of the role of HPV positive HNSCC versus HPV negative HNSCC. There is much in this manuscript that suggests that the authors are only concerned with the HPV negative disease. However, nowhere do they make this clear. It is critical that this be addressed right at the beginning, including the manuscript title. Otherwise it is very confusing. For example, in the introduction the authors state that "approx. 90% of head and neck cancers are squamous cell carcinomas which have a relatively high, approx. 50% five year mortality rate". This is true, but only for HPV negative HNSCC. The survival rate of HPV positive HNSCC is closer to 80-85%.

Our response: We are grateful for these comments, on the basis of which we have made extensive changes to the text, including the title. It is exactly the case that our data mainly consists of HPV/p16 negative patient samples (only 5.9% of all LIMA1 stained samples across all cohorts were p16 positive). The low p16 positivity across our cohorts can be explained that great majority of our samples are from other tumor sites than Oropharynx that is the site from where most of the HPV positive cancers are derived. We have clarified this message throughout in our revised manuscript, added a new table to the Appendix Data (Table EV3) describing p16 staining data from each of our study cohorts, and changed our title to better reflect the content and results of this work.

The title is needs to be revised. I would prefer something like "LIMA1 immunohistochemistry predicts curative intent surgery response in HPV negative head and neck cancer".

Our response: We fully agree with the Referee on this suggestion and have changed the title to the form requested by the Referee.

The first cohort of this study is a TMA comprised of patient material collected between 2005-2010 in Finland. Perhaps in Finland, as in North America the presence of HPV positive HNSCC was relatively low during those years. However, development of HPV positive HNSCC is growing at epidemic proportions around the world. I am not sure that the TMA is useful other than as an indicator of LIMA levels in HPVnegative disease.

Our response: This observation by the Referee is completely correct and we have been able to show in our still unpublished follow-up studies that also in Finland, the prevalence of p16/HPV positivity has increased significantly, especially in oropharyngeal SCC. However, as our prospective cohorts are very recent, this balances out the potential impact of the issue with the collection time of the TMA for the overall conclusions from the study. Due to the low HPV prevalence in our study data, we have changed the content and title of the manuscript to better reflect our results in HPV-negative HNSCC cancers.

The HNSCC cohort 2 that the authors employ is the head and neck TCGA database (n=528) however, there are 40 normal samples in the collection and 72 (or73) HPV positive samples. This is never mentioned or addressed, and this could be a confounding factor in their analyses. A re-analysis of the data is necessary with a separation and identification of HPV negative patients and HPV positive patients.

Our response: We have performed more comprehensive analyses of the HNSCC TCGA data in our revised manuscript, including the mRNA correlation between LIMA1 and HPV. The following new data have been added to our revised manuscript:

- Appendix Data Figure S3A, the correlation between LIMA1 and HPV mRNA expression.
- Appendix Data Figure S3B-C, the correlation between LIMA1 exomes and HPV expression.
- Ln. 553, Figure EV4, the correlation between LIMA1 and EMT genes.

In these analyses, expression from normal samples in TCGA HNSCC data were also taken into account.

The authors state that from their original cohort of 476 patients, they identified 312 cases that did not have any signs of metastasis based on, I assume, clinical reports (which is not stated but should be). "97 patients of this cohort were treated with curative intended surgery". 32 patients died during the 5 years of follow up (these don't add up to the 128 patients available in the cohort (line 199)). did these 32 receive chemo-radiation if they

were surgery alone?

And did the screen of the TMA from which this data was derived support this separation? Meaning did the 97 that had surgery alone and were still alive at 5 year FU have low or undetectable LIMA1 and the 32 that died have high LIMA1? If this was stated, then I missed it and apologize. Otherwise it should be examined.

Our response:

- *Standard practice of staging HNSCC included at the time of the cohort an upper panendoscopy and head and neck contrast CT as well as wider imaging studies when indicated. This question is now addressed in the materials&methods section (In 147-149): "Metastasis at the time of diagnose were ruled out by careful clinical investigation, upper panendoscopy, and pertinent imaging (mostly contrast CT), as per national guidelines applied at the time of cohort entry."*
- *The patients treated with surgery (n=97) and the patients dying during follow-up (n=32) are overlapping categories and the mention of these two numbers (in separate sentences) is merely to illustrate the representativeness of the cohort in relation to our research question. We have clarified this ambiguity by removing the first sentence, as we justify as well in our response below.*
- *The association between adjuvant treatments and LIMA1 expression is a very interesting issue, which we have also tried to investigate. A challenge from a statistical point of view has been the qualitative and temporal variation in the adjuvant treatments given in the patients in nmHNSCC cohort; some nmHNSCC patients have received, for example, RT treatment after recurrence or metastasis, some have received CRT treatment and some new surgery and adjuvant treatments. This is why we ended up, in our previous manuscript version to use the expression '97 patients... with curative intended surgery' and did not use '97 that had surgery alone'-impression. However, we find this sentence confusing and decided to remove the entire sentence from this new manuscript version. It is clearer to simply state that 'Notably, 25% (n=32) of the patients with nmHNSCC died of HNSCC cancer during 5 years of follow-up.' However, in our ongoing studies, we have also sought to further clarify the associations between LIMA1 expression and various adjuvant therapies, and we hope to shed more light on these issues in our future publications.*

Line 261-162 "...prognostic role of LIMA1 could be extended to other cancers types..." which would be useful but the only other cancer type the authors examined was pancreatic cancer. I think that a more types should be examined before I accept that LIMA1 would be widely used.

Our response: *It is true that our own results are limited only to HNSCC and pancreatic cancer. However, recent studies strongly support that LIMA1 play a significant role in regulating cancer aggressiveness also in cancers other than HNSCC (PMID: 39076043, PMID: 25569716, doi: <https://doi.org/10.1101/2024.06.27.600789>). In these publications, the aggressiveness-regulating significance of LIMA1 has been demonstrated in cholangiocarcinoma, lymphoma and breast cancer. In our revised manuscript, we have added and referenced also these references in the Discussion section (In 650-652).*

The data in Figure 2B, C is intriguing; however, the cohort is too small for any general claims.

Our response: *We fully agree with the referee regarding the small size of this subcohort. The main role of this cohort was to technically confirm the role of LIMA1 isoforms in the*

detection of HNSCC aggressiveness and to optimize LIMA1 antibodies to match Western blot results. The results are however fully supported by the results from the second prospective study both demonstrating that none of the LIMA1alpha low patients died of HNSCC during follow-up.

I applaud the authors efforts and look forward to seeing this work published.

Our response: *We are very grateful to the Reviewer for these very encouraging greetings.*

21st Mar 2025

Dear Prof. Westermarck,

Thank you for the submission of your revised manuscript to EMBO Molecular Medicine. I am pleased to inform you that we will be able to accept your manuscript pending the following final amendments:

- 1) Authors: We note name discrepancy in the manuscript and in our submission system: Linda Nissi vs. Liisa Nissi. Please correct.
- 2) Figures:
 - During a standard image analysis, we detected potential aberrations in the figure set, and we would like to clarify these issues before accepting your manuscript for publication. We kindly invite you to check western blot image in Figure 4G that does not match to the corresponding source data image. Please clarify and correct. Also, please make sure that all figures are accurate.
 - Please remove all figures from the manuscript and upload the EV Figures as individual, high-resolution files. Figure legends and EV figure legends should be moved to the end of the main manuscript file.
 - We note that Figure 2 is 195 MB big. If possible, please upload a smaller-sized figure.
- 3) In the main manuscript file, please do the following:
 - Please address all comments suggested by our data editors listed below:
 - o Figure legends:
 1. Please note that the exact p values are not provided in the legends of figures 2D, 4K, EV2 A, EV4 B, S1 A, S5 D, F.
 2. Please indicate the statistical test used for data analysis in the legends of figures 1B, 2B, 4B, C, E, F, H, I, K; EV4 B, S1 A.
 3. Please note that in figures 4K, S5 D, F there is a mismatch between the annotated p values in the figure legend and the annotated p values in the figure file that should be corrected.
 4. Please note that the box plots need to be defined in terms of minima, maxima, centre, bounds of box and whiskers, and percentile in the legends of figures 4B, C, E, F, H, I, K; S3 A-C.
 5. Please note that information related to n is missing in the legends of figures 1C, S1 A; S3 A-C.
 6. Please note that the error bars are not defined in the legends of figures 1C, S1 A.
 7. Please note that the scale bar needs to be defined for figure S4
 8. Please note that scale bar and its definition are missing for figures 1B, 3B, C, E; EV1 C, H; S1 B, D, E, G, H; S2 A, B, D, E, G, H, J, K; S4 A, S5 A, B.
 9. Please note that the white arrows are not defined in the legend of figure S4 E. This needs to be rectified.
 - Correct order and titles of manuscript sections: Abstract / Keywords / The Paper Explained / Introduction / Results / Discussion / Methods / Data Availability / Acknowledgements / Disclosure and Competing Interests Statement / References / Main Figure Legends / Tables / Expanded View Figure Legends.
 - Add callouts for Figure 3D.
 - Rename "Conflict of interests" to "Disclosure Statement & Competing Interests". We updated our journal's competing interests policy in January 2022 and request authors to consider both actual and perceived competing interests. Please review the policy <https://www.embopress.org/competing-interests> and update your competing interests if necessary.
 - Indicate in legends exact n and exact p values, not a range, along with the statistical test used. To keep the figures "clear" some authors found providing an Appendix table Sx with all exact p-values preferable. You are welcome to do this if you want to.
 - In Methods, please include statement that the experiments with patient samples conformed to the principles set out in the WMA Declaration of Helsinki and the Department of Health and Human Services Belmont Report.
 - Please include structured Methods section that includes a Reagents and Tools Table (should be uploaded as a separate file) followed by a Methods and Protocols section. More information on how to adhere to this format as well as downloadable templates (.docx) for the Reagents and Tools Table can be found in our author guidelines: <https://www.embopress.org/page/journal/17574684/authorguide#structuredmethods>
- An example of a paper with Structured Methods can be found here: <https://www.embopress.org/doi/full/10.1038/s44320-024-00037-6#sec-4>
- Correct the reference citation in the text and reference list. In the text, a reference should be cited by author and year of publication. Include a space between a word and the opening parenthesis of the reference that follows. In the reference list, citations should be listed in alphabetical order. Where there are more than 10 authors on a paper, 10 will be listed, followed by "et al.". Please check "Author Guidelines" for more information. <https://www.embopress.org/page/journal/17574684/authorguide#referencesformat>
- 4) Appendix: Add a table of contents with page numbers to the first page. Please remove "The remark checklist" and move all supplementary methods to the main "Methods" section. Rename EV Tables to "Appendix Table S1" etc. and place them after Appendix Figures. Please update their callouts in the main manuscript file.
- 5) Funding: Please make sure that information about all sources of funding are complete in both our submission system and in the manuscript. Currently, project numbers and Business Finland Research to Business Funding (Diary number:1278/31/2021) are missing in our system. Please correct.
- 6) Synopsis:
 - Synopsis image: Please resize the image to 550 px-wide x (300-600)-px high and upload it as a high-resolution jpeg file.

7) As part of the EMBO Publications transparent editorial process initiative (see our Editorial at <http://embomolmed.embopress.org/content/2/9/329>), EMBO Molecular Medicine will publish online a Review Process File (RPF) to accompany accepted manuscripts. This file will be published in conjunction with your paper and will include the anonymous referee reports, your point-by-point response and all pertinent correspondence relating to the manuscript. Let us know whether you agree with the publication of the RPF and as here, if you want to remove or not any figures from it prior to publication. Please note that the Authors checklist will be published at the end of the RPF.

8) Please provide a point-by-point letter INCLUDING my comments as well as the reviewer's reports and your detailed responses (as Word file).

I look forward to reading a new revised version of your manuscript as soon as possible.

Yours sincerely,

Zeljko Durdevic

*** IMPORTANT INFORMATION ***

- 1) a .doc formatted version of the manuscript text (including Figure legends and tables)
- 2) Separate figure files
- 3) a letter INCLUDING the reviewer's reports and your detailed responses to their comments.

Also, and to save some time should your paper be accepted, please read below for additional information regarding some features of our research articles:

- 1) Glossary: EMBO Molecular Medicine articles will be accompanied by a glossary explaining some of the terms used for laymen. I identified the following:

_____, _____, _____

Could you please help us in identifying terms that may need an "explanation" other terms that we can add to the glossary.

- 2) For more information: This is a short list of related web links for further consultation by the readers. Could you identify some relevant ones? Examples are patient associations, OMIM related links, databases, authors websites, etc.

- 3) Pending issues: At the end of each article we will have a box highlighting issues that still need further studies and where research efforts should converge (we call this the Pending issues box). From my reading I would say:

but I can see there may be many more. Could you work on this as well?

- 4) Disclosure and competing interest statement: Please include a statement declaring any competing commercial interests in relation to your submitted work.

- 5) Please note that we now mandate that all corresponding authors list an ORCID digital identifier. This takes <90 seconds to complete. We encourage all authors to supply an ORCID identifier, which will be linked to their name for unambiguous name identification.

Currently, our records indicate that the ORCID for your account is 0000-0001-7478-3018.

Link Not Available

-

Thank you,

Zeljko Durdevic

***** Reviewer's comments *****

Referee #1 (Comments on Novelty/Model System for Author):

There are no current prognostic markers for curative intent surgery for clinically diagnosed non-metastatic HNSCC. There is a pressing need to identify IHC appropriate markers that will differentiate between patients with excellent prognosis with surgery alone and those that will likely need systemic therapy in addition to surgery.

Referee #1 (Remarks for Author):

Qiao have taken my initial review comments to heart and done a thorough job addressing them. In my view, the new data related to LIMA1 isoform expression, HPV status, and correlation with EMT signatures has strengthened the manuscript. They have also addressed my minor corrections, improving the overall quality of the submission.

Referee #2 (Comments on Novelty/Model System for Author):

I have always believed that this was a potentially useful clinical approach to de-escalating treatment of HPV- HNSCC,

Referee #2 (Remarks for Author):

I am happy with the revised manuscript and thank the authors for their efforts.

LIMA1-alpha staining predicts curative intent surgery response in HPV negative head and neck cancer

Qiao et al.,

Response to editorial comments:

We were very delighted to notice that both reviewers clearly appreciated our efforts in revising the manuscript based on their comments and that the manuscript is suitable to be accepted for EMM following technical revisions. As described below, we have now addressed all the editorial comments. We thank the editorial team for very efficient processing of our manuscript and hope that the work is now suitable to be published at *Embo Molecular Medicine*.

1. Authors: We note name discrepancy in the manuscript and in our submission system: Linda Nissi vs. Liisa Nissi.

Our response: It is now updated in the system.

2. Figures:

- During a standard image analysis, we detected potential aberrations in the figure set, and we would like to clarify these issues before accepting your manuscript for publication. We kindly invite you to check western blot image in Figure 4G that does not match to the corresponding source data image. Please clarify and correct. Also, please make sure that all figures are accurate.

Our response: Thank you for noticing that. They were the same results from different experiments which we mistook as the source file of the blot. However now we have updated the matching western blot in the source data file.

- Please remove all figures from the manuscript and upload the EV Figures as individual, high-resolution files. Figure legends and EV figure legends should be moved to the end of the main manuscript file.

Our response: All the figures and EV figures are removed from the manuscript file as suggested and the legends has been moved to the end of the main manuscript file.

- We note that Figure 2 is 195 MB big. If possible, please upload a smaller-sized figure.

Our response: This figure contains 23 individual IHC images which explains the big size. The small IHC figures in the panel 2A are already so small that if we decrease the resolution of the original images there is a great risk that they become uninformative. We would therefore ask a permission to retain the original size of this figure.

3. In the main manuscript file, please do the following:

- Please address all comments suggested by our data editors listed below:
 - o Figure legends:

1. Please note that the exact p values are not provided in the legends of figures 2D, 4K, EV2 A, EV4 B, S1 A, S5 D, F.

Our response: All the figures has the p values indicated in it so the p values were added in the legends as follows. "The Exact p values were as indicated". Since all the exact p values are in the figures, we used this statement in the legends. This has been done by referring other EMM publications.

2. Please indicate the statistical test used for data analysis in the legends of figures 1B, 2B, 4B, C, E, F, H, I, K; EV4 B, S1 A.

Our response: All the statistical tests can now be found in the figure legends.

3. Please note that in figures 4K, S5 D, F there is a mismatch between the annotated p values in the figure legend and the annotated p values in the figure file that should be corrected.

Our response: Thank you mentioning it is now corrected

4. Please note that the box plots need to be defined in terms of minima, maxima, centre, bounds of box and whiskers, and percentile in the legends of figures 4B, C, E, F, H, I, K; S3 A-C.

Our response: All the box mentioned box plots now has been defined in the legends.

5. Please note that information related to n is missing in the legends of figures 1C, S1 A; S3 A-C. **Our response:** The missing n values are now updated in the figure legends. The number of samples included for 1C, S1A clinical studies is 312 and the samples for each IHC stains vary and this information can be found in the supplementary table.

6. Please note that the error bars are not defined in the legends of figures 1C, S1 A.

Our response: The error bars are 95% Confidence intervals which is now mentioned in the legends.

7. Please note that the scale bar needs to be defined for figure S4.

8. Please note that scale bar and its definition are missing for figures 1B, 3B, C, E; EV1 C, H; S1 B, D, E, G, H; S2 A, B, D, E, G, H, J, K; S4 A, S5 A, B.

Our response to both points 6 and 7: All the missing scale bar are now in the figure and legends.

9. Please note that the white arrows are not defined in the legend of figure S4 E. This needs to be rectified.

Our response: White arrows indicate invading cells. This is now defined in the legend.

- Correct order and titles of manuscript sections: Abstract / Keywords / The Paper Explained / Introduction / Results / Discussion / Methods / Data Availability / Acknowledgements / Disclosure and Competing Interests Statement / References / Main Figure Legends / Tables / Expanded View Figure Legends.

Our response: the main manuscript file is in the above-mentioned order now.

- Add callouts for Figure 3D.

Our response: The callout was 3C instead of 3D which is fixed now.

- Rename "Conflict of interests" to "Disclosure Statement & Competing Interests". We updated our journal's competing interests policy in January 2022 and request authors to consider both actual and perceived competing interests. Please review the policy <https://www.embopress.org/competing-interests> and update your competing interests if necessary.

Our response: It is now renamed.

- Indicate in legends exact n and exact p values, not a range, along with the statistical test used. To keep the figures "clear" some authors found providing an Appendix table Sx with all exact p-values preferable. You are welcome to do this if you want to.

Our response: Yes all the exact n and p values can now be found in the figures and legends.

- In Methods, please include statement that the experiments with patient samples conformed to the principles set out in the WMA Declaration of Helsinki and the Department of Health and Human Services Belmont Report.

Our response: This statement is now included.

- Please include structured Methods section that includes a Reagents and Tools Table (should be uploaded as a separate file) followed by a Methods and Protocols section. More information on how to adhere to this format as well as downloadable templates (.docx) for the Reagents and Tools Table can be found in our author

guidelines: <https://www.embopress.org/page/journal/17574684/authorguide#structuredmethods>

An example of a paper with Structured Methods can be found here: <https://www.embopress.org/doi/full/10.1038/s44320-024-00037-6#sec-4>

Our response: We now provide a Tool and reagents table.

- Correct the reference citation in the text and reference list. In the text, a reference should be cited by author and year of publication. Include a space between a word and the opening parenthesis of the reference that follows. In the reference list, citations should be listed in alphabetical order. Where there are more than 10 authors on a paper, 10 will be listed, followed by "et al.". Please check "Author Guidelines" for more information. <https://www.embopress.org/page/journal/17574684/authorguide#referencesformat>

Our response: We have now used the suggested reference format.

4. Appendix: Add a table of contents with page numbers to the first page. Please remove "The remark checklist" and move all supplementary methods to the main "Methods" section. Rename EV Tables to "Appendix Table S1" etc. and place them after Appendix Figures. Please update their callouts in the main manuscript file.

Our response: This is done.

5. Funding:

Please make sure that information about all sources of funding are complete in both our submission system and in the manuscript. Currently, project numbers and Business Finland Research to Business Funding (Diary number:1278/31/2021) are missing in our system. Please correct.

Our response: This is corrected now.

6. Synopsis:

- Synopsis image: Please resize the image to 550 px-wide x (300-600)-px high and upload it as a high-resolution jpeg file.
- Please check your synopsis text and image before submission with your revised manuscript. Please be aware that in the proof stage minor corrections only are allowed (e.g., typos).

Our response: This is done.

7. As part of the EMBO Publications transparent editorial process initiative (see our Editorial at <http://embomolmed.embopress.org/content/2/9/329>), EMBO Molecular Medicine will publish online a Review Process File (RPF) to accompany accepted manuscripts. This file will be published in conjunction with your paper and will include the anonymous referee reports, your point-by-point response and all pertinent correspondence relating to the manuscript. Let us know whether you agree with the publication of the RPF and as here, if you want to remove or not any figures from it prior to publication. Please note that the Authors checklist will be published at the end of the RPF.

8. Please provide a point-by-point letter INCLUDING my comments as well as the reviewer's reports and your detailed responses (as Word file).

Our response: Yes.

16th Jun 2025

Dear Prof. Westermarck,

We are pleased to inform you that your manuscript is accepted for publication and is now being sent to our publisher to be included in the next available issue of EMBO Molecular Medicine.
